# Layered roles of *fruitless* isoforms in specification and function of male aggression-promoting neurons in *Drosophila*

**Margot Wohl[1,2†], Kenichi Ishii[1,2†], Kenta Asahina[1]\***

[1]Molecular Neurobiology Laboratory, The Salk Institute for Biological Studies, La Jolla, United States; [2]Neuroscience Graduate Program, University of California, San Diego, United States

**Abstract** Inter-male aggressive behavior is a prominent sexually dimorphic behavior. Neural circuits that underlie aggressive behavior are therefore likely under the control of sex-determining genes. However, the neurogenetic mechanism that generates sex-specific aggressive behavior remains largely unknown. Here, we found that a neuronal class specified by one of the *Drosophila* sex determining genes, *fruitless* (*fru*), belongs to the neural circuit that generates male-type aggressive behavior. This neuronal class can promote aggressive behavior independent of another sex determining gene, *doublesex* (*dsx*), although *dsx* is involved in ensuring that aggressive behavior is performed only toward males. We also found that three *fru* isoforms with different DNA binding domains show a division of labor on male aggressive behaviors. A dominant role of *fru* in specifying sex-specific aggressive behavior may underscore a genetic mechanism that allows male-type aggressive behavior to evolve at least partially independently from courtship behavior, which is under different selective pressures.

**\*For correspondence:**
kasahina@salk.edu

[†]These authors contributed equally to this work

**Competing interests:** The authors declare that no competing interests exist.

## Introduction

Aggression is widely observed across animal species, especially in the context of reproductive activities (*Huntingford and Turner, 1987*). It is mostly males that perform agonistic behaviors, which are often ritualized, to gain access to mating partners. Some species have even developed male-specific weaponry organs to gain advantage (*Rico-Guevara and Hurme, 2018*). Although these inter-male aggressive behaviors must have evolved to increase fitness, the effect on sexual selection is likely indirect. Unlike courtship behavior, which is selected directly through female preference, females can choose only the end result of inter-male contest and not the aggressive behavior itself. Another distinction of aggressive behavior from courtship behavior is that it can be exhibited by both males and females, but the motor programs are often sexually dimorphic (*Hashikawa et al., 2018*; *Nilsen et al., 2004*). Moreover, males are aggressive almost exclusively towards other conspecific males, and not towards females. In other words, aggressive behavior is sexually dimorphic, and its execution is strongly influenced by the sex of the target animal.

The sexually dimorphic nature of aggressive behavior predicts the existence of sexually dimorphic neuronal circuits that support this behavior. Both in mice and flies, mutations in genes with sexually dimorphic expression patterns, including sex-determining genes or steroid sex hormone receptors, can alter aggressive behaviors (*Juntti et al., 2010*; *Ogawa et al., 2000*; *Vrontou et al., 2006*; *Wu et al., 2009*; *Xu et al., 2012*). Neurons expressing sex-determining genes also influence aggressive behaviors (*Asahina et al., 2014*; *Chan and Kravitz, 2007*; *Deutsch et al., 2020*; *Hashikawa et al., 2017*; *Hoopfer et al., 2015*; *Koganezawa et al., 2016*; *Lee et al., 2014*;

*Palavicino-Maggio et al., 2019*; *Yang et al., 2013*), which is consistent with the above prediction that male-type aggression and female-type aggression may be generated through distinct neural circuits. However, sexual dimorphisms within neurons or circuits that are causally responsible for the generation of sexually dimorphic behavior remain unclear. The genetic and neuronal mechanisms which ensure a male only executes aggressive behaviors towards a male target is not well understood either. Although sex-specific sensory cues seem to play critical roles for both mice (*Chamero et al., 2007*; *Hattori et al., 2016*; *Stowers et al., 2002*) and flies (*Fernández et al., 2010*; *Wang and Anderson, 2010*), how the sensory information affects the operation of circuits that generate sexually dimorphic aggressive actions is largely uncharacterized.

Sexual dimorphism of aggressive behavior in the common fruit fly *Drosophila melanogaster* has been well documented. Males perform aggressive behaviors almost exclusively to one another with a higher intensity than inter-female aggressive behaviors (*Chen et al., 2002*; *Nilsen et al., 2004*; *Vrontou et al., 2006*). Attack actions are also dimorphic: a male fly often perform lunges, in which it raises its front legs and pounces down toward a target fly (*Chen et al., 2002*; *Dow and von Schilcher, 1975*; *Hoyer et al., 2008*; *Jacobs, 1960*), whereas a female fly performs headbutts, in which it quickly moves its body horizontally and hits a target with its head (*Nilsen et al., 2004*; *Ueda and Kidokoro, 2002*). Sex of *Drosophila* is determined by two genes that encode transcription factors, *doublesex* (*dsx*) and *fruitless* (*fru*). These two genes undergo sex-specific splicing to produce male-specific (dsxM, fruM) and female-specific (dsxF, fruF) transcripts (*Yamamoto and Koganezawa, 2013*). *fru* further undergoes another round of alternative splicing among at least 3 different zing finger DNA binding domains (*Usui-Aoki et al., 2000*). *dsx* and *fru* collectively specify anatomical and behavioral sexual dimorphisms (*Marín and Baker, 1998*; *Yamamoto and Koganezawa, 2013*). In *Drosophila*, several classes of sexually dimorphic neurons, which express *dsx* or *fru* (or both), are implicated in aggressive behaviors (*Asahina et al., 2014*; *Chan and Kravitz, 2007*; *Deutsch et al., 2020*; *Hoopfer et al., 2015*; *Koganezawa et al., 2016*; *Palavicino-Maggio et al., 2019*). An outstanding question is which of these neurons are critical for the execution of male-type or female-type aggressive behaviors. Results we present in *Ishii et al. (2020)* support the hypothesis that *dsx* specifies an execution mechanism for male courtship behavior, while *fru* specifies neural circuits underlying sexually dimorphic aggressive actions (*Vrontou et al., 2006*). Currently, however, it remains unclear which, if any, of the *fru*-expressing neurons constitutes the execution mechanism for male-type aggressive behavior.

Here, we characterized Tk-GAL4$^{FruM}$ neurons (*Asahina et al., 2014*) as a part of a fruM-dependent neural circuit that generates male-type aggressive behavior. Tk-GAL4$^{FruM}$ neurons are *fru*-expressing, male-specific neurons that are not only capable of promoting male-type aggressive behavior, but also are necessary for spontaneous male aggression (*Asahina et al., 2014*). We first found that Tk-GAL4$^{FruM}$ neurons are specified solely by *fru*. We then demonstrated that the activation of Tk-GAL4$^{FruM}$ neurons elicited male-type aggressive behaviors irrespective of *dsx* genotype. However, *dsx* has a role in refining the application specificity of male-type aggressive behaviors toward a male target fly. We also characterized the impact of different *fru* isoforms on male aggression. We found that only one of the three *fru* isoforms is necessary for the specification of Tk-GAL4$^{FruM}$ neurons, while another isoform is necessary to perform spontaneous aggressive behaviors. The dominant role of *fru* in specifying the circuit for executing sex-specific aggressive behavior is in contrast to the importance of *dsx* for the execution mechanism for courtship behavior. The distinct roles of *dsx* and *fru* on the specification of courtship- and aggression-controlling neurons can be the genetic basis underlying evolution of beneficial behaviors for inter-male contests while preserving species-specificity for courtship behavior.

## Results

### fruM specifies aggression-promoting Tk-GAL4$^{FruM}$ neurons

Tk-GAL4$^{FruM}$ neurons are male-specific *fru*-expressing neurons that not only promote male-type aggressive behavior upon activation, but also are necessary for normal levels of male aggression (*Asahina et al., 2014*). We first characterized the behavioral impact of optogenetic activation of these neurons in the presence of male or female target flies (*Figure 1A*), because sex of the target fly can be an important biological variable in determining the function of social behavior-controlling

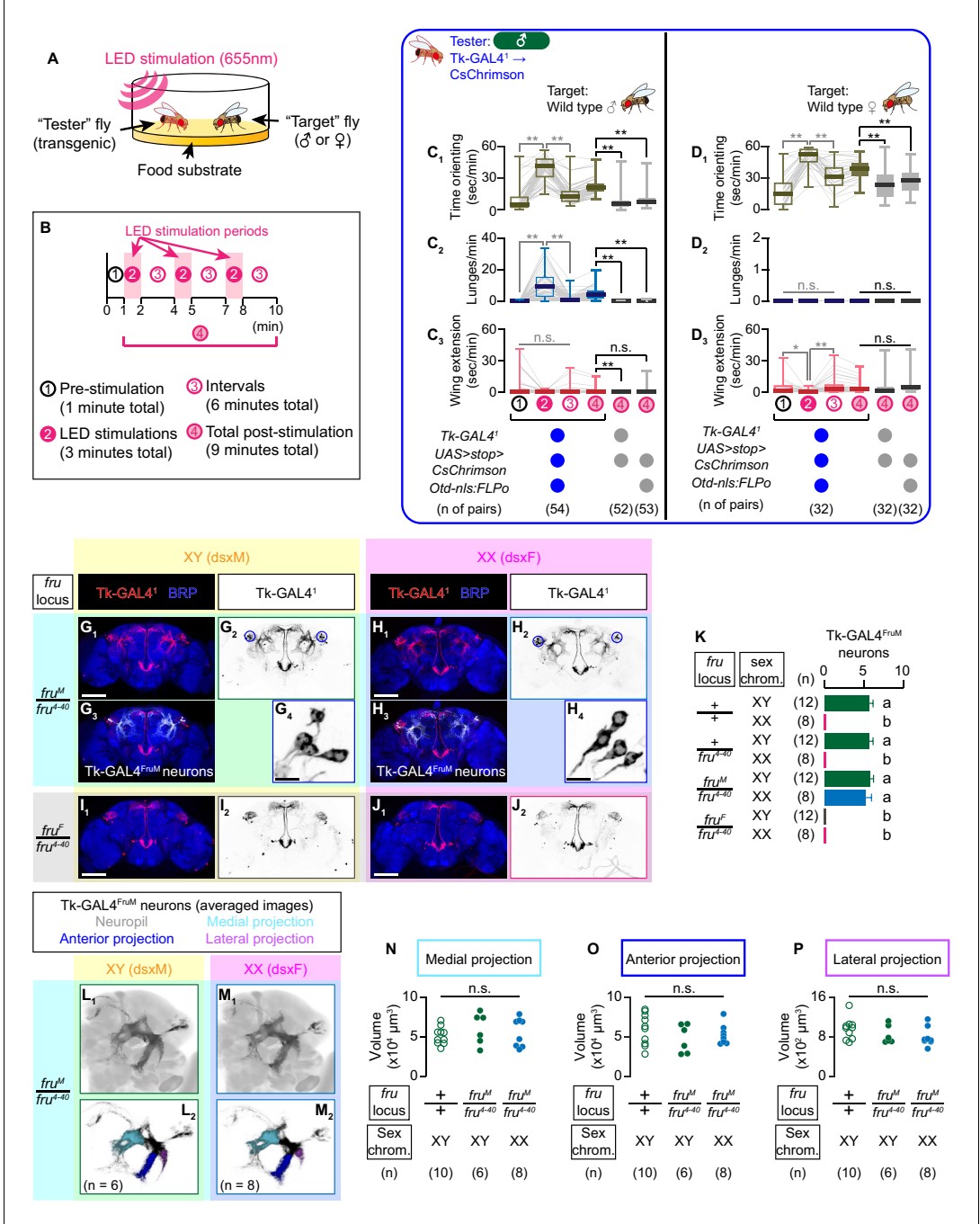

**Figure 1.** Tk-GAL4[FruM] neurons are specified by *fru*. (A) Schematic of the design of behavioral assays. (B) Schematic of the optogenetic stimulation paradigm. Time windows 1–4 represent periods in which behavioral parameters are pooled and calculated in subsequent panels. (C, D) Boxplots of time orienting ($C_1$, $D_1$), lunges ($C_2$, $D_2$), and wing extensions ($C_3$, $D_3$) by the tester flies during the time windows 1–4 (see B). Their genotypes and pair numbers are indicated below the plots. Gray lines represent single testers. Target flies are either group-housed wild-type males (C) or mated females (D). In gray: **$p<0.01$, *$p<0.05$, n.s. $p>0.05$ (Kruskal-Wallis one-way ANOVA and post-hoc Wilcoxon signed rank test). In black: **$p<0.01$, n.s. $p>0.05$ (Kruskal-Wallis one-way ANOVA and post-hoc Mann-Whitney U-test). (E) Schematics of the sex-determination pathway in *Drosophila*. (F) Schematic of the four sex genotypes defined by *dsx* and *fru* splicing. (G–J) Expression of CsChrimson:tdTomato under the control of *Tk-GAL4[1]* and *Otd-nls:FLPo* (red in $G_1$–$J_1$, black in $G_{2,3}$-$J_{2,3}$) in brains of a male (G), fruM female (H), fruF male (I), and female (J) flies is visualized together with a neuropil marker BRP (blue) by immunohistochemistry. Traced Tk-GAL4[FruM] neurons in a male ($G_3$) and a fruM female ($H_3$) are shown in white. Circle: soma (right cluster is enlarged in $G_4$ and $H_4$). Scale bar: 100 μm ($G_1$–$J_1$), 10 μm ($G_4$, $H_4$). (K) Mean number of cell bodies per hemibrain visualized by anti-DsRed antibody in each genotype represented in G–J and *Figure 1—figure supplement 2J–M*. Error bars: S.D. Genotypes and number of hemibrains examined are indicated to the left. Lowercase letters denote significance group ($p<0.01$, one-way ANOVA with post-hoc Tukey's honestly significant difference test).

*Figure 1 continued on next page*

*Figure 1 continued*

(**L, M**) Z-projection of segmented, registered, and averaged images of CsChrimson:tdTomato expression under the control of $Tk\text{-}GAL4^1$ and *Otd-nls: FLPo* (black) in the standard *Drosophila* brain (gray in $L_1$, ($M_1$). Number of used hemibrains are indicated in $L_2$, $M_2$. N-P: Volumes of medial projection (**N**), anterior projection (**O**), and lateral projection (**P**) of Tk-GAL4$^{FruM}$ neurons in males with genotypes indicated below. Their genotypes and pair numbers are indicated below the plots. n.s. $p > 0.05$ (Kruskal-Wallis one-way ANOVA).

The online version of this article includes the following figure supplement(s) for figure 1:

**Figure supplement 1.** Effects of *fru* locus genotypes on the specification and function of Tk-GAL4$^{FruM}$ neurons.

**Figure supplement 2.** Detailed characterization of Tk-GAL4$^{FruM}$ neurons.

**Figure supplement 3.** Effects of *fru* alleles on the specification of Tk-GAL4$^{FruM}$ neurons.

neurons. We quantified the time in which a tester fly orients toward a target fly, number of lunges, and duration of unilateral wing extensions (shorthanded as 'wing extension' hereafter) using automated behavior classifiers before, during, and after the optogenetic stimulation of Tk-GAL4$^{FruM}$ neurons with programmed LEDs (*Figure 1B*; see also Materials and methods).

The tester flies in which CsChrimson was expressed under the control of $Tk\text{-}GAL4^1$ within the brain (which phenocopies the manipulation of Tk-GAL4$^{FruM}$ neurons; *Asahina et al., 2014*) showed robust aggression toward the target male flies, mainly during LED stimulation (*Figure 1* $C_2$, *Figure 1—figure supplement 1A2*; *Watanabe et al., 2017*). This coincides with increased time orienting toward the target fly (*Figure 1$C_1$*, *Figure 1—figure supplement 1A1*), but wing extension toward males was not affected (*Figure 1$C_3$*, *Figure 1—figure supplement 1A3*; see also *Video 1* – Part 1). These behavioral effects were qualitatively similar across different stimulation frequencies (*Figure 1—figure supplement 1C*). In contrast, we observed no lunges toward female target flies after the same optogenetic manipulations (*Figure 1$D_2$*, *Figure 1—figure supplement 1B2*). The tester flies increased the time orienting toward the target female during LED stimulations (*Figure 1$D_1$*, *Figure 1—figure supplement 1B1*; see also *Video 1* – Part 2), suggesting that the activation of Tk-GAL4$^{FruM}$ neurons can promote interaction regardless of the sex of the target fly. Neither did we observe lunges when $Tk\text{-}GAL4^1$-labeled neurons were optogenetically activated in the absence of any target fly (*Figure 1—figure supplement 1D*). Thus, the activation of Tk-GAL4$^{FruM}$ neurons induces lunges strictly when a male target fly is present. In addition, we observed that tester flies performed less wing extensions toward target females during LED stimulation than during ISIs (*Figure 1$D_3$*). However, the total amount of wing extensions was not significantly different from those of the two genetic controls. These phenotypes are largely consistent with those observed when Tk-GAL4$^{FruM}$ neurons were thermogenetically activated (*Asahina et al., 2014*), and reinforces the idea that the sex of the target fly impacts the behavioral outcome of manipulations of Tk-GAL4$^{FruM}$ neuronal activity.

As was previously shown (*Asahina et al., 2014*), Tk-GAL4$^{FruM}$ neurons are specified only in males (*Figure 1G,J*). We next addressed the role of *dsx* and *fru* (*Figure 1—figure supplement 2A*) on specifying Tk-GAL4$^{FruM}$ neurons. Two splicing mutations for *fru*, $fru^M$ and $fru^F$ (*Demir and Dickson, 2005*), forces male-type (fruM) and female-type (fruF) splicing of *fru* transcripts irrespective of the chromosomal sex composition, which normally determines the sex-determination genetic cascade. As a result, these mutations allowed us to create 'fruF males' (expressing dsxM and fruF) and 'fruM females' (expressing dsxF and fruM), thus dissociating contributions of *dsx* and *fru* on sexual

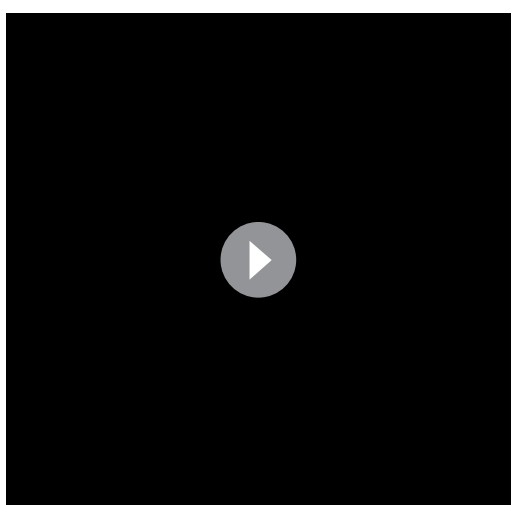

**Video 1.** Representative behavior of a male tester fly that expresses CsChrimson:tdTomato under the control of $Tk\text{-}GAL4^1$ and *Otd-nls:FLPo* toward a wild-type male (Part 1) or a wild-type female (Part 2) target fly, at the onset and offset of LED stimulation. In Part 2, the tester male was not actively performing any behavior toward the target female when the LED turned on.
https://elifesciences.org/articles/52702#video1

dimorphism at the organismal level (*Figure 1—figure supplement 2B*). fruM females are reported to show lunges and other male-type aggressive actions (*Vrontou et al., 2006*), suggesting that *fru* is the chief architect of neural circuits underlying sexually dimorphism in aggressive behavior. However, two *fru*-expressing, aggression-promoting neurons (male-type NP2631 ∩ dsx$^{FLP}$ neurons (*Koganezawa et al., 2016*) and P1$^a$ neurons *Hoopfer et al., 2015*) are not specified in this genotype (see *Ishii et al., 2020*). These observations strongly suggest that not all *fru*-expressing neurons underlie the sexual dimorphism of aggressive actions, even if the activation of such neurons in a normal male induces male-type aggressive behavior. Other neuronal populations specified by *fru*, but not *dsx*, must constitute a neural substrate for the execution mechanism for male-type aggression.

In contrast to male-type NP2631 ∩ dsx$^{FLP}$ neurons or P1$^a$ neurons, Tk-GAL4$^{FruM}$ neurons were present in fruM females (*Figure 1H*), but not in fruF males (*Figure 1I*). Consistent with this observation, we detected expression of FruM, but not Dsx, proteins in Tk-GAL4$^{FruM}$ neurons (*Figure 1—figure supplement 2C,D*). The cell body number of Tk-GAL4$^{FruM}$ neurons in fruM females was indistinguishable from that in males (*Figure 1K*). To compare Tk-GAL4$^{FruM}$ neuronal morphology in males and fruM females, we first traced Tk-GAL4$^{FruM}$ neurons using a volume visualization software FluoRender (see Materials and methods for details). The distinct neuronal processes of Tk-GAL4$^{FruM}$ neurons allowed us to segment them in an unambiguous manner (*Figure 1G$_3$, H$_3$*). We then registered the original and segmented confocal images of brains to a unisex standard brain (*Bogovic et al., 2018*) using non-rigid spatial transformation (*Jefferis et al., 2007*). The three-dimension average of genetically isolated Tk-GAL4$^{FruM}$ neurons (*Figure 1—figure supplement 2E–G*; *Asahina et al., 2014*) almost perfectly overlapped with the average of segmented Tk-GAL4$^{FruM}$ neurons (*Figure 1—figure supplement 2I, Video 2*), confirming the accuracy of the segmentation. The averaged morphology of registered Tk-GAL4$^{FruM}$ neurons from males (*Figure 1L*) and from fruM females (*Figure 1M*) appeared virtually identical (*Video 3*). To confirm this initial observation, we further segmented three prominent neurites that emanate from the lateral junction (*Yu et al., 2010*) of Tk-GAL4$^{FruM}$ neurons (*Figure 1L$_2$, M$_2$*). None of these three processes showed significant difference in volume among *fru +/+* control males, *fru$^M$/fru$^{4-40}$* males, and fruM females (*Figure 1N–P*). While we acknowledge that this approach might miss finer scale differences in neuroanatomy, we conclude that Tk-GAL4$^{FruM}$ neurons are specified predominantly by *fru. dsx* and *fru* therefore have clearly separable roles for the specification of neuronal populations that are important for *Drosophila* aggressive behaviors.

Presence of the *fru$^{4-40}$* allele did not affect cell number or overall morphology of Tk-GAL4$^{FruM}$ neurons (*Figure 1K*, *Figure 1—figure supplement 2F–M*). Consistent with this, the activation of *Tk-GAL4$^1$*-labeled neurons triggered similar behavioral changes in the backgrounds of +/+, +/fru$^{4-40}$, or fru$^M$/fru$^{4-40}$ males (*Figure 1—figure supplement 2N*). However, heterozygosity of *fru$^M$* resulted in incomplete specification of Tk-GAL4$^{FruM}$ neurons in a chromosomally female background (*Figure 1—figure supplement 3A–C*). We also found that *fru$^{FLP}$*, a knock-in allele of *fru* that expresses a DNA recombinase Flippase (*Yu et al., 2010*), is hypomorphic (*Figure 1—figure supplement 3D,E*), raising the necessity to use caution when attempting to transform *Drosophila* sex by manipulating the *fru* locus.

## Tk-GAL4$^{FruM}$ neurons are a part of a fruM-dependent circuit for the execution of male-type aggressive behavior

We next investigated the function of Tk-GAL4$^{FruM}$ neurons in fruM females. Specifically, we wished to obtain insights into the sex recognition mechanism and execution mechanism in the context of aggression. Unfortunately, male 'target' flies often perform vigorous courtship toward female tester flies (*Figure 2—figure supplement 1B*), which interferes with tester females' social behavior (namely headbutts) towards target males (*Figure 2—figure*

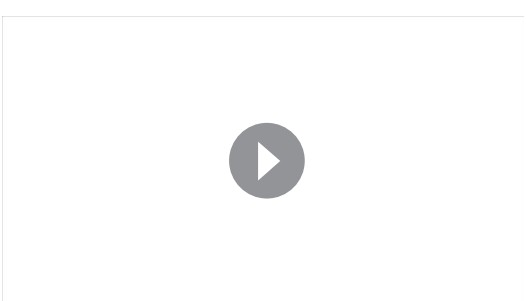

**Video 2.** 3D-rendered average image of segmented and registered Tk-GAL4$^{FruM}$ neurons (green) and registered Tk-GAL4$^{FruM}$ neurons that are genetically isolated (blue). Gray represents the standard unisex *Drosophila* brain (*Bogovic et al., 2018*).
https://elifesciences.org/articles/52702#video2

supplement 1A; *Vrontou et al., 2006*). Following a previous study (*Vrontou et al., 2006*), we used group-housed fruF males (which are defective at enhancing courtship toward females; *Demir and Dickson, 2005*; *Pan and Baker, 2014*; *Villella et al., 1997*; *Figure 2—figure supplement 1B*) instead of wild-type male flies as targets when using anatomically female (dsxF) flies as testers.

Optogenetic activation of the *Tk-GAL4$^1$*-labeled neurons in fruM females induced lunges toward a fruF male target for the majority of testers (*Figure 2A$_2$*, *Figure 2—figure supplement 2A2*; see also *Video 4* – Part 1). Although orienting time increased during LED stimulations compared to during ISIs (*Figure 2A$_1$*, *Figure 2—figure supplement 2A1*), no overall change in the amount of wing extensions was observed compared to genetic controls (*Figure 2A$_3$*). This indicates that Tk-GAL4$^{FruM}$ neurons are part of a fruM-dependent, but *dsx*-independent, neuronal circuit for male-type aggressive behavior. The motion sequence underlying the lunges observed in this genotype were indistinguishable from wild-type male lunges (*Video 5*).

However, we noted two differences compared to the same manipulation in male flies. First, lunges were not induced in fruM females as consistently as in males. Forty-two percent (17/40) of fruM females did not perform a single lunge after LED stimulation (*Figure 2C*). In contrast, over 96% of male testers lunged toward a male target (*Figure 2C*). The median of induced lunges observed in fruM females (2) was lower than observed for male testers (58). Second, fruM females occasionally lunged toward a female target (*Video 4* – Part 2), in addition to a fruF male target. Even though lunges were induced even less frequently than toward fruF males (*Figure 2A$_2$, B$_2$, C*, *Figure 2—figure supplement 2A2*, 2B$_2$), it is still noteworthy because male testers seldom lunge toward a female target, even after optogenetic activation of the *Tk-GAL4$^1$* neurons (*Figure 1D$_2$*, *Figure 2C*). We noticed that headbutts performed by the female target toward these testers often outnumbered the lunges performed by the tester toward the target (*Figure 2—figure supplement 3B*). This was not the case when the target flies were fruF males (*Figure 2—figure supplement 3A*). These counter-attacks by female targets may have contributed to the lower number of lunges toward female targets than toward fruF male targets.

We asked whether social isolation, which elevates levels of aggression in males (*Hoffmann, 1990*; *Wang et al., 2008*), could enhance the induction of lunges. Interestingly, optogenetic activation of *Tk-GAL4$^1$*-labeled neurons in single-housed fruM females failed to induce a single lunge toward fruF male targets (*Figure 2—figure supplement 3C2* F$_2$). Instead, this manipulation resulted in higher levels of orientation and wing extensions than was exhibited by group-housed flies (*Figure 2—figure supplement 3C1*, C$_3$). These data suggest that the relatively high levels of courtship behavior seen in fruM females interferes with the aggression-promoting function of Tk-GAL4$^{FruM}$ neurons. Consistent with this idea, group-housed fruM females that did not lunge toward a target fly after optogenetic stimulation of *Tk-GAL4$^1$*-labeled neurons tended to show higher levels of wing extensions than testers who lunged (*Figure 2—figure supplement 3E*, F$_1$). Lastly, optogenetic activation of Tk-GAL4$^{FruM}$ neurons did not induce headbutts (*Figure 2—figure supplement 2A3*, B$_3$), confirming that this genotype is using a male-type aggressive action exclusively (*Vrontou et al., 2006*).

Overall, the results above delineate complementary roles of *dsx* and *fru* on male aggressive behavior. The activation of Tk-GAL4$^{FruM}$ neurons induces lunges even in the absence of *dsx*-dependent aggression-promoting neurons. Tk-GAL4$^{FruM}$ neurons are therefore part of a fruM-dependent execution mechanism for male aggressive behavior (*Figure 2D*). However, the reduced intensity and compromised target sex selectivity of lunges shown by fruM females suggest that a *dsx*-dependent mechanism may play a role in establishing a target sex-selective application of aggressive behavior. This mechanism can inhibit the execution of lunges specifically toward females, which ensures that lunges are performed exclusively by males towards other males (*Figure 2D*), or can enhance aggression specifically toward male targets indirectly by inhibiting courtship toward them.

## Female-type NP2631 ∩ dsx$^{FLP}$ neurons in fruM females do not promote aggression

Having established that Tk-GAL4$^{FruM}$ neurons can trigger lunges independent of *dsx* splicing pattern, we wondered whether other sexually dimorphic neurons are involved in the execution of male-type aggressive behaviors in the fruM female brain. One candidate was the NP2631 ∩ dsx$^{FLP}$ neuron group, which was reported to promote aggressive behaviors in both males and females (*Koganezawa et al., 2016*).

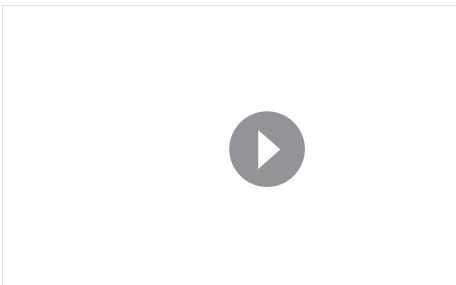

**Video 3.** 3D-rendered average image of registered Tk-GAL4^FruM neurons in male (green) and in fruM female (blue). Gray represents the standard unisex *Drosophila* brain (**Bogovic et al., 2018**).
https://elifesciences.org/articles/52702#video3

As was discussed above, *dsx* specifies the sexual dimorphism of NP2631 ∩ dsx^FLP neurons, such that the morphology of these neurons in fruM females is largely female-type, and not male-type (**Figure 3A**; see also **Ishii et al., 2020**). We confirmed that optogenetic activation of NP2631 ∩ dsx^FLP neurons in females robustly increased time orienting (**Figure 3C₁**, **Figure 3—figure supplement 1B1**) and headbutts (**Figure 3C₂**, **Figure 3—figure supplement 1B2**) toward female target flies, mostly during LED stimulation (see also **Video 6** – Part 1). This behavioral phenotype was consistently observed when the *fru* locus was either +/+, +/fru^{4-40} or fru^F/fru^{4-40}, similar to the case in males (**Figure 3—figure supplement 1C**). Interestingly, we observed similar behaviors toward the fruF male targets as well (**Figure 3B**, **Figure 3—figure supplement 1A**; see also **Video 6** – Part 2). Aggression-promoting neurons that belong to the *dsx*-expressing 'pC1' cluster have been also characterized in the female brain by using different genetic reagents (**Deutsch et al., 2020**; **Palavicino-Maggio et al., 2019**). Whether these neurons and the NP2631 ∩ dsx^FLP neurons we manipulated belong to the same subtypes remains undetermined. However, we found that NP2631 ∩ dsx^FLP neurons did not overlap with neurons labeled by the *R71G01-LexA* transgene (**Figure 3—figure supplement 1D,E**). This is consistent with recent findings that female 'pC1' neurons are anatomically and functionally heterogeneous (**Deutsch et al., 2020**; **Wang et al., 2020**; **Wu et al., 2019**).

The aggression-promoting function of NP2631 ∩ dsx^FLP neurons raises an intriguing question: do female-type NP2631 ∩ dsx^FLP neurons promote aggression in fruM females, and if so, are male-type or female-type aggressive behaviors induced (**Figure 3A**)? If fruM's role in defining sexual dimorphism in aggression prevails, activation of NP2631 ∩ dsx^FLP neurons in fruM may promote male-type aggression (lunges) even though these neurons exhibit female-type morphology. If the morphology of NP2631 ∩ dsx^FLP neurons dictates the sexual dimorphic action of aggressive behavior, the same manipulation may promote female-type aggressive actions (headbutts).

Interestingly, we did not observe induction of either lunges or headbutts (**Figure 3D2** and E₂,₃) when we optogenetically activated the NP2631 ∩ dsx^FLP neurons in fruM females. Thus, female-type NP2631 ∩ dsx^FLP neurons can promote headbutts only if other mechanisms that rely on sexually dimorphic splicing of *fru* are present in the brain. In addition, this result suggests that NP2631 ∩ dsx^FLP neurons with female-like morphology fail to establish functional connections with neurons that are part of the fruM-dependent execution mechanism for male-type aggression (which includes Tk-GAL4^FruM neurons; **Figure 3A**).

We therefore conclude that female-like NP2631 ∩ dsx^FLP neurons cannot participate in generation of aggressive behavior (e.g. headbutts) without separate populations of neurons that are specified in the absence of fruM. This finding raises a possibility that the sexual dimorphism of NP2631 ∩ dsx^FLP neurons, which is largely specified by *dsx*, does not determine the sexual dimorphism of aggressive actions. It is also possible that activities of both *dsx* and *fru* may be required for the complete functional transformation of NP2631 ∩ dsx^FLP neurons. As is shown in **Ishii et al. (2020)**, male-type NP2631 ∩ dsx^FLP neurons cannot trigger lunges without a fruM-dependent mechanism, underscoring the dominant role of *fru* and *fru*-dependent neural circuits in specifying sexually dimorphic action patterns during aggressive interactions.

Differential roles of *fru* isoforms on male-male interactions and specification of Tk-GAL4^FruM neurons fruM undergoes another layer of alternative splicing to create three transcript isoforms, fruMA, fruMB, and fruMC, (**Figure 4A**), which encode different zinc finger domains (**Usui-Aoki et al., 2000**). The loss of each of three fruM isoforms is known to affect male courtship behaviors in distinct manners (**Neville et al., 2014**; **von Philipsborn et al., 2014**; **Figure 4—figure supplement 1A,B**). After finding that fruM is important for specification of Tk-GAL4^FruM neurons, as well as for its capacity to

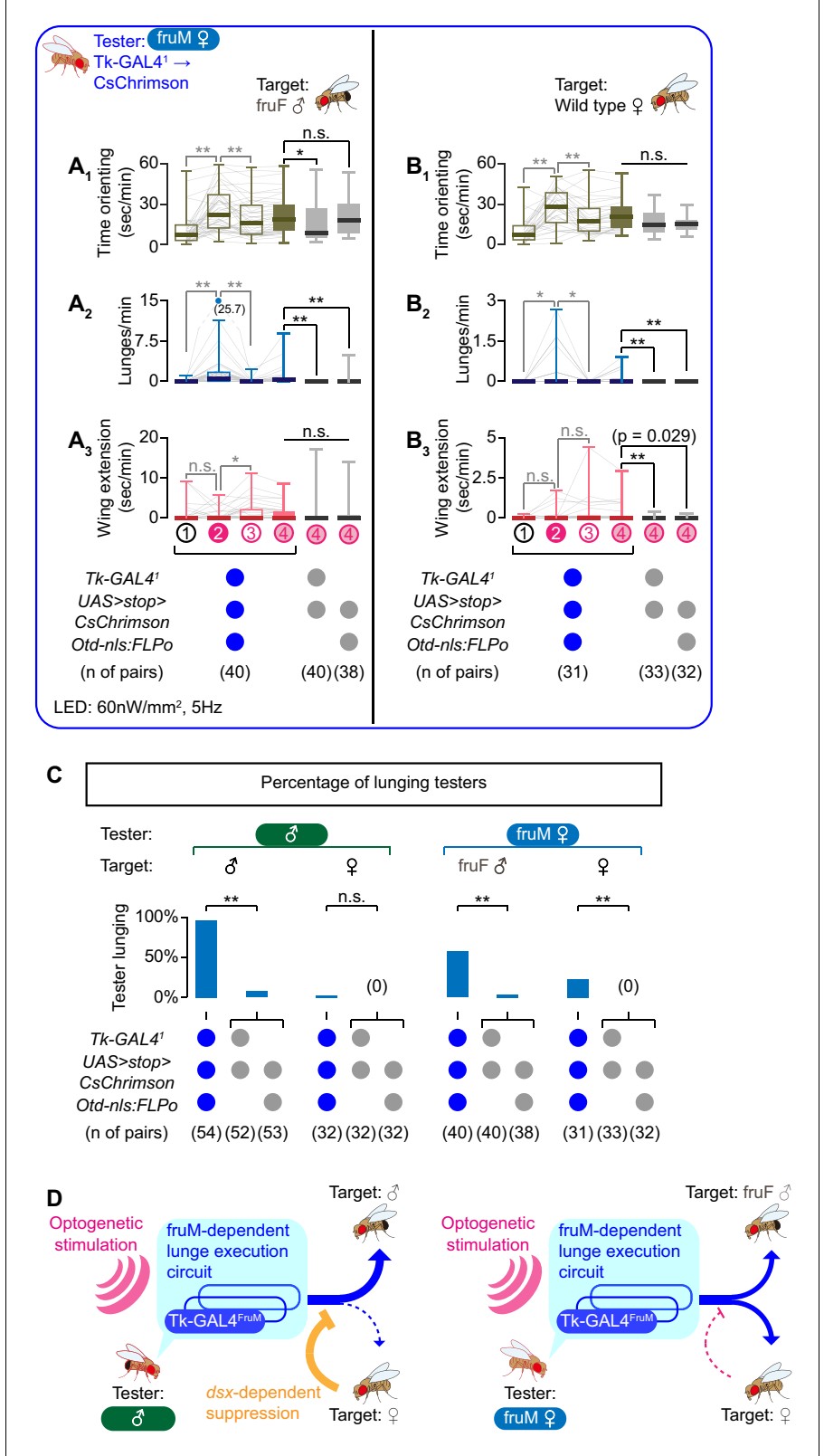

**Figure 2.** Tk-GAL4[FruM] neurons in fruM females promote male-type aggressive behaviors. (**A, B**) Boxplots of time orienting ($A_1$, $B_1$), lunges (A2, B2), and wing extensions ($A_3$, $B_3$) by the tester flies during the time windows 1–4 (see *Figure 1B*). Their genotypes and pair numbers are indicated below the plots. Gray lines represent single testers. Target flies are either group-housed fruF males (**A**) or wild-type mated females (**B**). In gray: **p<0.01, *p<0.05, n.s.

*Figure 2 continued on next page*

*Figure 2 continued*

p>0.05 (Kruskal-Wallis one-way ANOVA and post-hoc Wilcoxon signed rank test). In black: **p<0.01, *p<0.05, n.s. p>0.05 (Kruskal-Wallis one-way ANOVA and post-hoc Mann-Whitney U-test). (C) Ratio of male (dataset from *Figure 1C, D*) and fruM female (dataset from A), (B) tester flies that lunged toward each of the two target sexes (above) in the time window 4. Testers' genotypes and pair numbers are indicated below the plots. **p<0.01, n.s. p>0.05 (Fisher's exact test, two genetic controls are pooled). (D) Schematic summary of the roles of sex-determining genes for Tk-GAL4$^{FruM}$ neurons and male-type aggressive behavior.

The online version of this article includes the following figure supplement(s) for figure 2:

**Figure supplement 1.** Male target flies interfere with female tester flies' behaviors.
**Figure supplement 2.** Behaviors induced by optogenetic stimulation of Tk-GAL4$^{FruM}$ neurons in fruM females.
**Figure supplement 3.** Additional characterization of the function of Tk-GAL4$^{FruM}$ neurons in fruM females.

execute male-type aggressive actions, we next addressed whether different isoforms of fruM have distinct roles for the specification and function of Tk-GAL4$^{FruM}$ neurons.

First, we characterized how isoform-specific mutations affect spontaneous male-type aggressive behavior after social isolation (*Hoffmann, 1990*; *Wang et al., 2008*). In parallel with their previously characterized differential effects on courtship (*Neville et al., 2014*; *von Philipsborn et al., 2014*), we found defects of male aggressive behavior in isoform-specific mutants. Loss of fruMA caused no reduction of lunges compared to the *fru$^{4-40}$* heterozygous genetic control (*Figure 4B$_2$*). In contrast, we observed a dramatic reduction in male-type aggressive behaviors by the loss of fruMB or fruMC isoforms (*Figure 4B$_2$*). While fruMC mutants showed decreased orientation time toward a male target, orienting times of fruMA and fruMB mutants were comparable to the *fru$^{4-40}$* heterozygous genetic control (*Figure 4A$_1$*). This reduction in lunges in fruMB mutants was qualitatively recapitulated when the *fru$^F$* allele was used to create trans-heterozygotes (*Figure 4C*), and when other mutations affecting FruB zinc finger domains (*von Philipsborn et al., 2014*) were tested in trans with *fru$^{4-40}$* (*Figure 4—figure supplement 1C*). We observed no consistent reduction in overall activity levels between fruMB mutants and genetic controls (*Figure 4—figure supplement 1D*), or in the total amount of sleep exhibited by these flies (*Figure 4—figure supplement 1E*). Male-male courtship was differentially affected in three mutants (*Figure 4A$_3$*), reflecting varied degrees of courtship defects toward females (*Figure 4—figure supplement 1A,B*; *Neville et al., 2014*; *von Philipsborn et al., 2014*). These data indicate that each of the three fruM isoforms make different contributions to aggressive behaviors. fruMA had a minimal impact on male aggressive behavior, whereas fruMB and fruMC were both necessary to maintain normal levels of male aggression. Moreover, the decreased aggression of fruMB mutants was not necessarily due to reduced activity levels or opportunities to interact with the target fly.

We next asked how each of the three fruM isoforms contributes to the specification of Tk-GAL4$^{FruM}$ neurons. In fruMC mutants, we found only $1.9 \pm 1.1$ (mean $\pm$ S.D., n = 8; *Figure 5D,E*) faintly labeled neurons in the brain area where Tk-GAL4$^{FruM}$ neurons are expected to appear, significantly fewer than the number of cells found in genetic controls ($5.6 \pm 0.5$, mean $\pm$ S.D., n = 8; *Figure 5A,E*). Weak staining of the Tk-GAL4$^{FruM}$ neurons in the fruMC mutants prevented us from visualizing their branching pattern. In contrast, fruMA and fruMB mutants did not affect the number of Tk-GAL4$^{FruM}$ neurons (*Figure 5E*). We compared the neuroanatomy of Tk-GAL4$^{FruM}$ neurons in these genotypes by registering immunohistochemically labeled brains on a standard brain. Tk-GAL4$^{FruM}$ neurons from

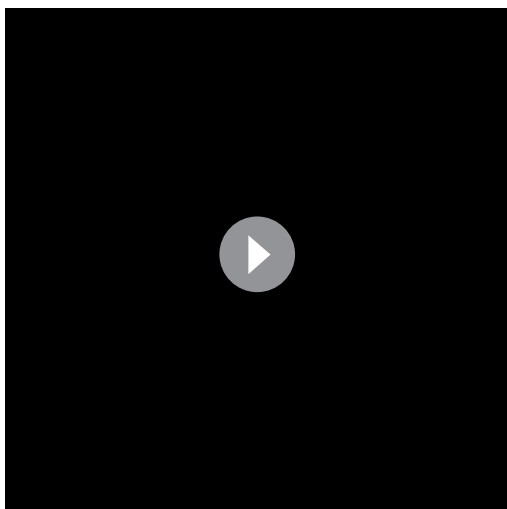

**Video 4.** Representative behavior of a fruM female tester fly that expresses CsChrimson:tdTomato under the control of *Tk-GAL4$^1$* and *Otd-nls:FLPo* toward a fruF male (Part 1) or a wild-type female (Part 2) target fly, at the onset and offset of LED stimulation.
https://elifesciences.org/articles/52702#video4

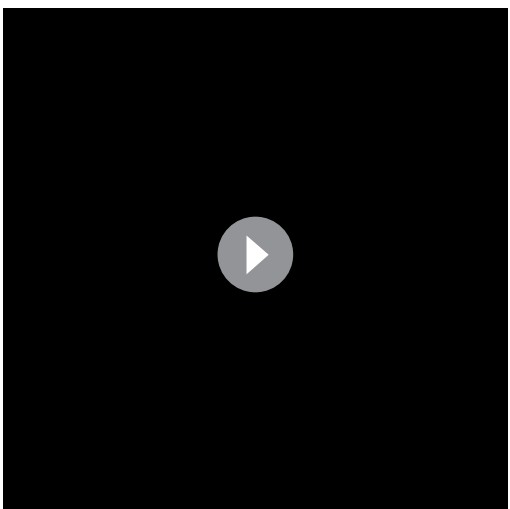

**Video 5.** A slow-motion comparison of a lunge executed by a male (top left) and fruM female (right) tester fly that express CsChrimson:tdTomato under the control of *Tk-GAL4¹* and *Otd-nls:FLPo*. The target fly is a wild-type male (top left), a fruF male (top right), and a wild-type female (bottom right), respectively.
https://elifesciences.org/articles/52702#video5

fruMA (*Figure 5G*) or fruMB (*Figure 5H*) mutants are indistinguishable from each other or from *fru⁴⁻⁴⁰* heterozygous control males (*Figure 5F*; see also *Video 7*). We measured the volumes of three prominent neurites that emanate from the lateral junction (*Yu et al., 2010*) of Tk-GAL4^FruM neurons (*Figure 5F2–H2*; see also *Figure 1L₁, M₁*), and detected no significant differences among *fru⁴⁻⁴⁰* heterozygous controls, fruMA mutants, and fruMB mutants (*Figure 5I–K*). Heterozygosity of all isoform-specific mutations did not change the cell body number or gross morphology of the Tk-GAL4^FruM neurons (*Figure 5E*, *Figure 5—figure supplement 1A–D*), either.

Taken together, we found that fruMA, fruMB and fruMC have distinct effects on male aggression and specification of the Tk-GAL4^FruM neurons. fruMA appeared largely dispensable for both, whereas fruMC was necessary for both. Interestingly, fruMB was necessary for normal levels of male-male aggression, but was not required for specification of the Tk-GAL4^FruM neurons.

## The Tk-GAL4^FruM neurons in fruMB mutants can induce male aggression

While the reduction of male aggression in fruMC mutants can be explained by defects in specification of the Tk-GAL4^FruM neurons, which are necessary for normal levels of aggression (*Asahina et al., 2014*), this same logic does not apply for the fruMB mutants, since these mutants have Tk-GAL4^FruM neurons that appear to retain their morphology. One possible explanation is that the Tk-GAL4^FruM neurons in fruMB mutants have neurophysiological defects and are no longer functional. To address this, we quantified the excitability of Tk-GAL4^FruM neurons in fruMB mutants by measuring their intracellular calcium response while optogenetically stimulating them (*Figure 6A,B*). The overall calcium response dynamics and magnitudes of Tk-GAL4^FruM neurons in fruMB mutants were comparable to those seen in the *fru⁴⁻⁴⁰* heterozygous controls (*Figure 6C,D*), suggesting that Tk-GAL4^FruM neurons in fruMB mutants are capable of physiologically responding to depolarizing stimuli.

Even though Tk-GAL4^FruM neurons exhibited normal excitability, it remains possible that neural circuits downstream of the Tk-GAL4^FruM do not respond to inputs from Tk-GAL4^FruM neurons in fruMB mutants, due to altered development or physiology of these downstream circuits, or inability of Tk-GAL4^FruM neurons to excite downstream neurons. To address this possibility, we optogenetically activated *Tk-GAL4¹*-labeled neurons in fruMA, fruMB, or fruMC mutants. Optogenetic activation of *Tk-GAL4¹*-labeled neurons in fruMB mutants elicited robust levels of male aggression across several LED intensities, as seen in genetic controls (*Figure 6E–G*, *Figure 6—figure supplement 1A–C*). This result suggests that Tk-GAL4^FruM neurons in fruMB mutants, once activated, can induce male aggression as efficiently as in the *fru* wild-type males, and argues against the hypothesis that the mutation of fruMB proteins causes defective development of downstream neurons or reduced efficiency of Tk-GAL4^FruM neurons to excite downstream neurons. Instead, these results favor the alternative hypothesis that fruMB mutation reduces aggression-promoting input to Tk-GAL4^FruM neurons from upstream circuits. This is likely mediated through fruM-dependent mechanisms as well, since activation of Tk-GAL4^FruM neurons can completely overcome the lack of such input.

Overall, our results show that three isoforms of fruM assume distinct roles in the specification and function of sexually dimorphic neurons that control male-type aggressive behaviors (*Figure 6H*). Namely, fruMC is necessary for the specification of Tk-GAL4^FruM neurons, whereas fruMB is necessary for a normal level of spontaneous aggression.

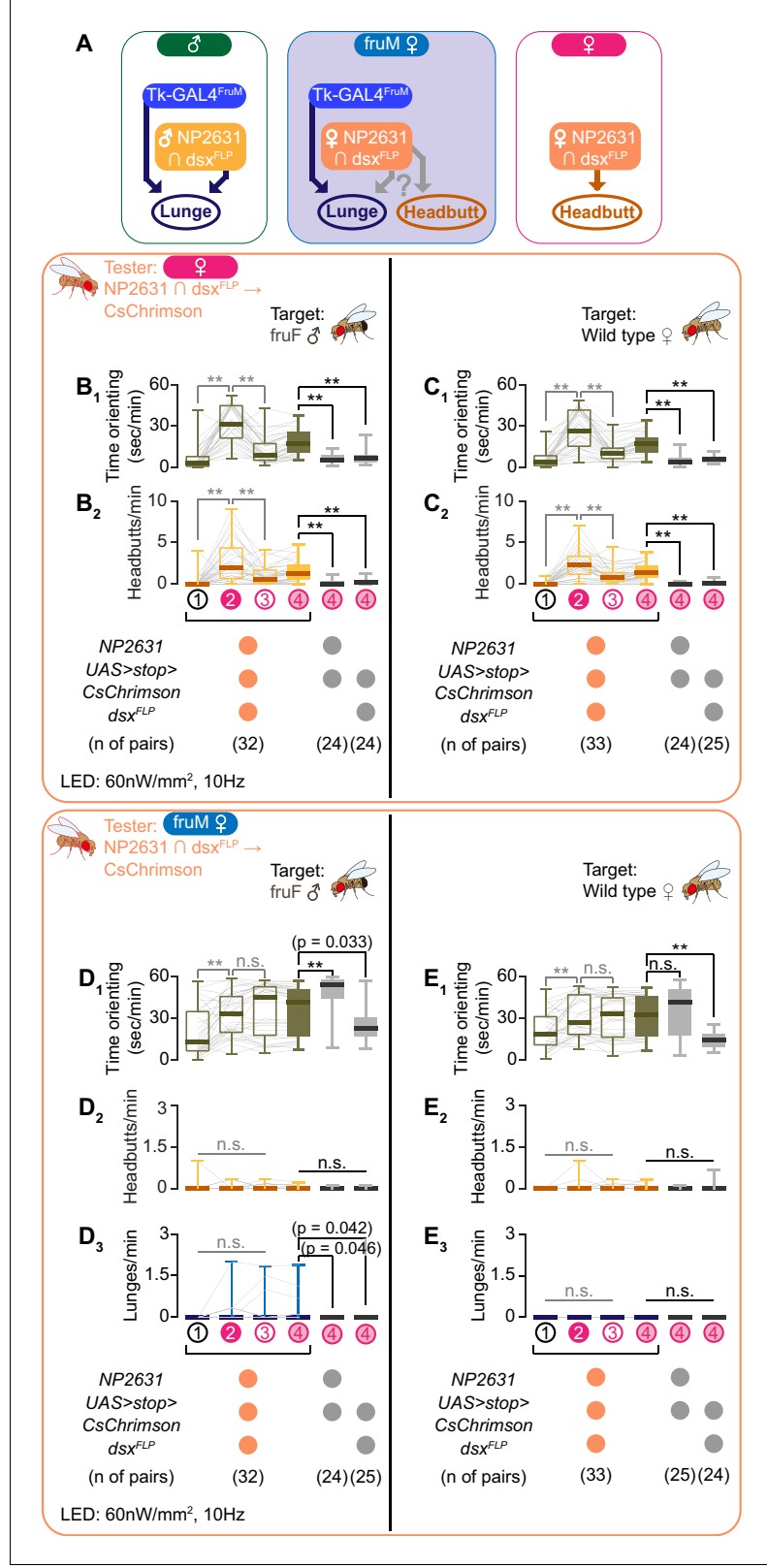

**Figure 3.** NP2631 $\bigcap$ dsx$^{FLP}$ neurons do not promote either male- or female-type aggressive behavior in fruM females. (**A**) Schematics of how Tk-GAL4$^{FruM}$ and NP2631 $\bigcap$ dsx$^{FLP}$ neurons (see *Ishii et al., 2020* for details) are specified in each genotype. Female-type NP2631 $\bigcap$ dsx$^{FLP}$ neurons are specified in fruM females, raising a question about how they contribute to aggression in this genotype. (**B–E**) Boxplots of time orienting (B$_1$–E$_1$), 

*Figure 3 continued on next page*

*Figure 3 continued*

headbutts ($B_2$–$E_2$), and lunges ($D_3$, $E_3$) by the female (**B**, **C**) or fruM female (**D**, **E**) tester flies during the time windows 1–4 (see *Figure 1B*). Their genotypes and pair numbers are indicated below the plots. Gray lines represent single testers. Target flies are either group-housed fruF males (**B**, **D**) or mated females (**C**, **E**). In gray: **p<0.01, n.s. p>0.05 (Kruskal-Wallis one-way ANOVA and post-hoc Wilcoxon signed rank test), In black: **p<0.01, *p<0.05, n.s. p>0.05 (Kruskal-Wallis one-way ANOVA and post-hoc Mann-Whitney U-test).

The online version of this article includes the following figure supplement(s) for figure 3:

**Figure supplement 1.** Behaviors induced by optogenetic activation of NP2631 $\bigcap$ dsx$^{FLP}$ neurons in females.

## Discussion

In this study, we identified Tk-GAL4$^{FruM}$ neurons as part of a fruM-dependent execution mechanism for male-type aggression. The exclusive role of fruM on the specification of a neural circuit for the execution of male-type aggression is in contrast to the circuit for the execution of male-type courtship behavior, for which *dsx* plays a major role (see *Ishii et al., 2020*). One role of *dsx* on male-type aggression appears to be to prevent inappropriate aggression toward a female target, which suggests that *dsx* may be involved in the sex-recognition mechanism in the context of agonistic interactions. Furthermore, we found that each of three fruM isoforms play separable roles on male-type aggressive behavior. Our finding reveals a layered genetic mechanism of the *fru* gene for the specification and function of a sexually dimorphic aggression-controlling circuit, which may provide a genetic substrate for the male-specific evolution of contest rituals.

### *fru* specifies a neural circuit for sexually dimorphic aggressive actions

We found that the optogenetic activation of Tk-GAL4$^{FruM}$ neurons in fruM female tester flies induced male-type aggressive behavior. This result indicates that at least Tk-GAL4$^{FruM}$ neurons and downstream populations form a neural circuit that is sufficient for the execution of male-type aggressive behavior. While this does not necessarily mean that fruF males or females lack the capability to execute male-type aggressive behavior, there has been no report that these two sex genotypes perform lunges spontaneously (*Vrontou et al., 2006*). Moreover, female-type NP2631 $\bigcap$ dsx$^{FLP}$ neurons fail to promote aggressive behaviors (lunges or headbutts) in fruM females. NP2631 $\bigcap$ dsx$^{FLP}$ neurons were previously proposed to be the 'aggression output' of a switch circuit that selects between courtship and aggression (*Koganezawa et al., 2016*). Since stimulation of NP2631 $\bigcap$ dsx$^{FLP}$ neurons in fruF males did not trigger lunges (see *Ishii et al., 2020*), at least sexual dimorphism in the neuroanatomy of NP2631 $\bigcap$ dsx$^{FLP}$ neurons does not correlate with the sexual dimorphism in aggressive motor programs. These observations favor a hypothesis that neurons specified by *fru*, which include Tk-GAL4$^{FruM}$ neurons, form the execution mechanism for male-type aggressive action, and NP2631 $\bigcap$ dsx$^{FLP}$ neurons serve as modulatory neurons that act on this *fru*-dependent execution mechanism. Sex-specific splicing of *fru* defines what type of aggressive behavior an animal performs, likely by specifying a mutually exclusive neural circuit that generates either male-type or female-type aggressive actions.

The definitive role of *fru* in specifying the execution mechanism for aggression does not mean that *dsx* is irrelevant. In fact, fruM females lunge against females as well as fruF males, suggesting that *dsx* is important for suppressing inappropriate aggressive behavior toward a female target. Currently, we do not know whether it is the

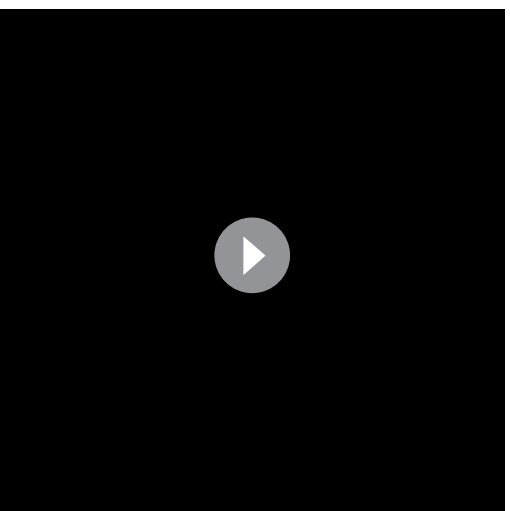

**Video 6.** Representative behavior of a female tester fly that expresses CsChrimson:tdTomato under the control of *NP2631* and *dsx$^{FLP}$* toward a wild-type female (Part 1) or a fruF male (Part 2) target fly, at the onset and offset of LED stimulation.
https://elifesciences.org/articles/52702#video6

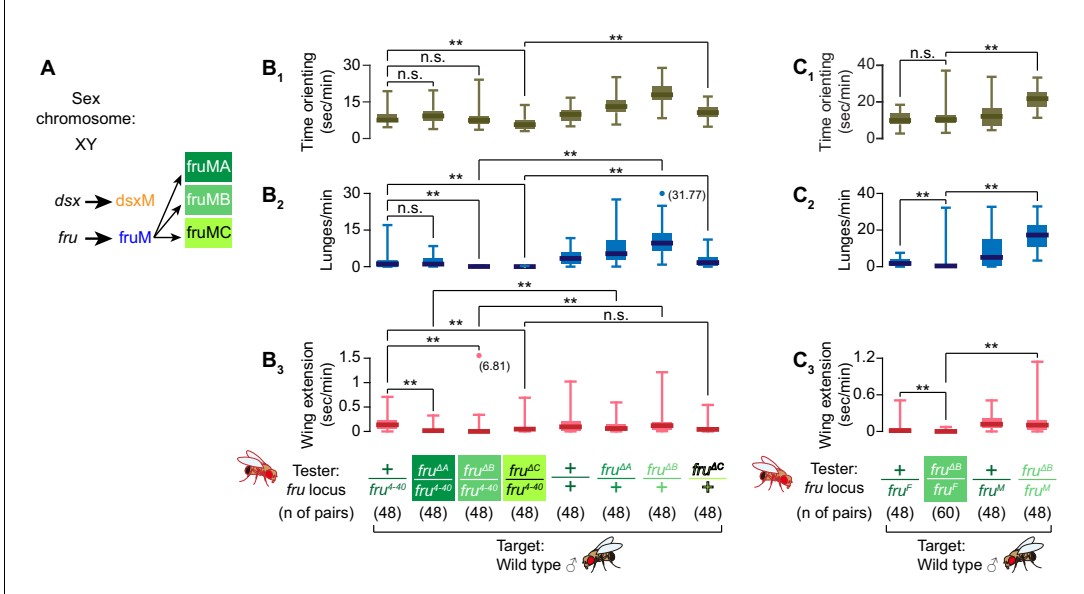

**Figure 4.** fruMB and fruMC are necessary for male aggression. (**A**) Schematics of *dsx* and *fru* splicing patterns in a male. (**B**, **C**) Boxplots of time orienting ($B_1$, $C_1$), lunges ($B_2$, $C_2$), and wing extensions ($B_3$, $C_3$) by the *fru* isoform-specific tester mutants (**B**) or fruMB tester mutants created in trans with the $fru^F$ allele (**C**), along with their genetic controls (single-housed for 6 days), toward group-housed wild-type target males during 30 min assays. Testers' genotypes of the *fru* locus and pair numbers are indicated below the plots. Data points that exceed the range (indicated left) are represented by dots, with exact values in parentheses. **p<0.01 (Mann-Whitney U-test with Bonferroni multiple comparison corrections).

The online version of this article includes the following figure supplement(s) for figure 4:

**Figure supplement 1.** Additional characterization of the isoform-specific *fru* mutations.

presence of dsxM, or the absence of dsxF, that is critical. Intriguingly, however, *dsx*-expressing neurons respond to sex-specific sensory cues, such as olfactory, gustatory, and auditory stimuli (*Deutsch et al., 2019*; *Zhou et al., 2014*; *Zhou et al., 2015*). Since thermogenetic activation of Tk-GAL4$^{FruM}$ neurons induces lunges toward a male target that lacks cuticular hydrocarbons (*Asahina et al., 2014*), one possible scenario is that a *dsx*-dependent circuit is responsible for detection or transmission of female-specific, aggression-suppressing cues that have yet to be identified. Alternatively, *dsx* in males may allow execution of male-type aggressive behavior specifically toward male targets indirectly by inhibiting courtship behavior toward males. fruM females that showed high levels of courtship behavior generally showed low levels of male-type aggression. This observation suggests that courtship behavior can interfere with aggressive behavior, possibly through feedforward inhibitory circuits (*Clowney et al., 2015*; *Kallman et al., 2015*; *Koganezawa et al., 2016*). Regardless of the mechanism, our results uncover distinct roles of *dsx* and *fru* on male-type aggressive behavior.

The contribution of *dsx* on the target sex-selective execution of aggressive behavior also implies that information about target sex can be supplied downstream of Tk-GAL4$^{FruM}$ neuronal activities and modify its behavioral influence. Currently, it remains unclear how the stimulation of Tk-GAL4$^{FruM}$ neurons is transformed into lunges, because the downstream partners of Tk-GAL4$^{FruM}$ neurons have not been characterized yet. It is possible that *dsx*-specified neurons serve as a 'block' against neuronal activity generated by Tk-GAL4$^{FruM}$ neurons and other unidentified constituents of the lunge execution mechanism. Since *fru*-expressing neurons have been comprehensively characterized in the male brain (*Cachero et al., 2010*; *Yu et al., 2010*), it may be necessary to establish an equally detailed catalog of *dsx*-expressing neurons for uncovering the neural mechanism by which male-type aggressive behavior is controlled in a target sex-dependent manner.

The neuronal population that supports execution of female-type aggressive behaviors has begun to be characterized only recently. Consistent with a previous observation (*Koganezawa et al., 2016*), we observed that NP2631 ∩ dsx$^{FLP}$ neurons in females promote female-type aggression. This population belongs to a cluster of *dsx*-expressing 'pC1' neurons, which is known to promote a

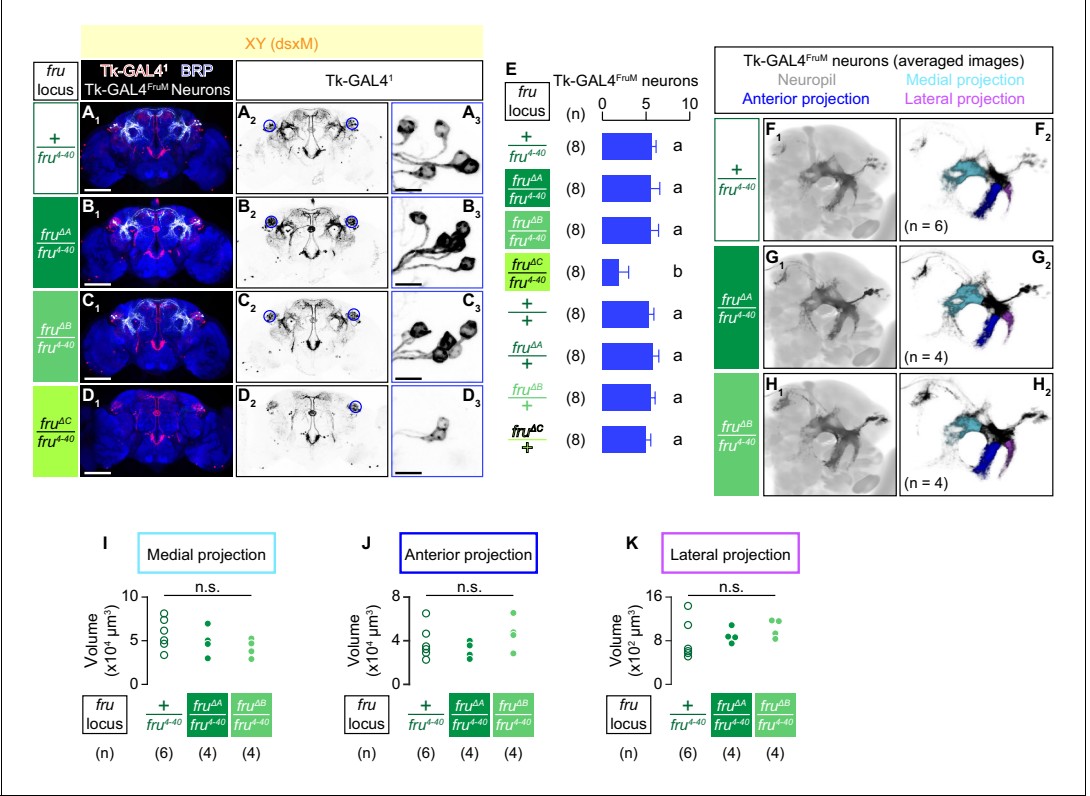

**Figure 5.** Only fruMC is necessary for the specification of Tk-GAL4<sup>FruM</sup> neurons. (A–D): Expression of CsChrimson:tdTomato under the control of *Tk-GAL4<sup>1</sup>* and *Otd-nls:FLPo* (red in A₁–D₁), black in A₂,₃-D₂,₃) in male brains is visualized together with a neuropil marker BRP (blue in A₁–D₁) by immunohistochemistry. Traced Tk-GAL4<sup>FruM</sup> neurons are shown in white. Circle: soma (right cluster is enlarged in A₃-D₃). Scale bar: 100 μm (A₁–D₁), 10 μm (A₃–D₃). (E) Mean number of cell bodies of Tk-GAL4<sup>FruM</sup> neurons per hemibrain is visualized by anti-DsRed antibody in each genotype represented in A–D and *Figure 5—figure supplement 1A–D*. Error bars, S.D. Lowercase letters denote significance group (p<0.01, one-way ANOVA with post-hoc Tukey's honestly significant difference test). (F–H) Z-projection of segmented, registered, and averaged images of CsChrimson:tdTomato expression under the control of *Tk-GAL4<sup>1</sup>* and *Otd-nls:FLPo* (black) in the standard *Drosophila* brain (gray in F₁–H₁). Number of used hemibrains are indicated in F₂–H₂. Medial projection (cyan), anterior projection (blue), and lateral projection (purple) are segmented and overlaid in F₂–H₂). For panels A–H, genotypes of the *fru* locus are indicated on the left. (I–K) Volumes of medial projection (I), anterior projection (J), and lateral projection (K) of Tk-GAL4<sup>FruM</sup> neurons in males. Their genotypes of the *fru* locus and pair numbers are indicated below the plots. n.s. p>0.05 (Kruskal-Wallis one-way ANOVA).

The online version of this article includes the following figure supplement(s) for figure 5:

**Figure supplement 1.** Additional characterization of the isoform-specific *fru* mutations.

variety of female social behaviors including aggression (*Deutsch et al., 2020*; *Palavicino-Maggio et al., 2019*; *Robie et al., 2017*), receptivity (*Deutsch et al., 2020*; *Zhou et al., 2014*) or even courtship-like behaviors (*Rezával et al., 2016*; *Wu et al., 2019*). In spite of the importance of 'pC1' neurons collectively on female social behaviors, female-type NP2631 ∩ dsx<sup>FLP</sup> neurons are unlikely to be a part of the execution mechanism that generates female-type aggressive actions. Our results suggest that there are uncharacterized sexually dimorphic neurons that are necessary for females to execute headbutts and other female-specific

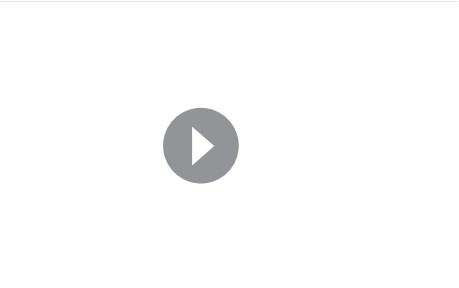

**Video 7.** 3D-rendered average image of registered Tk-GAL4<sup>FruM</sup> neurons in *fru<sup>4-40</sup>* heterozygous control male (dark green) and in fruMB mutant (*fru<sup>ΔB</sup>/fru<sup>4-40</sup>*) male (bright green). Gray represents the standard unisex *Drosophila* brain (*Bogovic et al., 2018*).
https://elifesciences.org/articles/52702#video7

aggressive actions, likely downstream of female-type NP2631 ∩ dsx$^{FLP}$ neurons.

Isoform-specific roles of *fru* on male-type aggression *fru* is a complex gene with multiple alternative splicing sites. Male-specific fruM transcripts can encode at least 3 different DNA binding domains (*Usui-Aoki et al., 2000*). These three isoforms have distinct impacts on gene expression (*Dalton et al., 2013*; *Neville et al., 2014*; *Vernes, 2015*), neuronal specification (*Billeter et al., 2006*; *Ito et al., 2016*; *Meissner et al., 2016*; *von Philipsborn et al., 2014*), and courtship behavior (*Neville et al., 2014*; *von Philipsborn et al., 2014*), prompting us to address whether these isoforms have differential roles on aggression. Indeed, we found that fruMC is important for the specification of Tk-GAL4$^{FruM}$ neurons, whereas fruMB is possibly important for perceiving or transmitting relevant sensory cues about male targets to an aggression execution mechanism. Because optogenetic stimulation of Tk-GAL4$^{FruM}$ neurons in fruMB mutants increases aggression as effectively as the same manipulation in normal males, any role of fruMB on modulating aggression is likely taking place upstream of Tk-GAL4$^{FruM}$ neurons. In order to understand the role of fruMB on the neural circuits controlling male-type aggression, it will be important to characterize the neural inputs to Tk-GAL4$^{FruM}$ neurons and to identify which of such neurons express fruMB (*von Philipsborn et al., 2014*).

Our result is in line with the previously reported importance of the fruMC isoform in the development of male-specific branching patterns in several fruM-expressing neurons (*Billeter et al., 2006*; *Ito et al., 2016*; *von Philipsborn et al., 2014*). Among them, activities of 'P1' neurons (*von Philipsborn et al., 2014*) and mAL/aDT2 neurons (*Ito et al., 2016*; *von Philipsborn et al., 2014*) are necessary for normal levels of courtship behavior (*Kallman et al., 2015*; *von Philipsborn et al., 2011*), reflecting the severe courtship defects of fruMC mutant males (*Neville et al., 2014*; *von Philipsborn et al., 2014*). Our observation that fruMC is necessary for both male aggression and specification of Tk-GAL4$^{FruM}$ neurons reveals its importance for male-type aggressive behavior as well, at behavioral and circuitry levels. Using an analogy from the function of steroid hormones in mammals (*McCarthy, 2008*), fruMC can be regarded as exerting 'organizational' functions, while fruMB may exert 'activation' functions, in the sense that fruMB is dispensable for the specification of a lunge execution mechanism, yet is necessary for induction of spontaneous aggressive behaviors. The additional support of *dsx* ensures the aggressive behaviors are applied specifically toward male targets.

The segregation of roles within *fru* isoforms may increase the gene's flexibility: for instance, change in the circuit connectivity underlying perception of male-specific cues can be independently altered if different isoforms have different roles in the aggression circuit. In mice, there are at least 3 different estrogen receptor genes (*Wu et al., 2009*). Although the specific function of each receptor on neural circuits that control sexual behaviors remains largely uncharacterized, it is tempting to speculate that a similar type of division of labor may exist among the estrogen receptor genes. For both *fru* and estrogen receptors, the functional characterization of genes which each isoform regulate will be the first important step to understand how the division of labor is implemented at the molecular level.

## *fru* may allow males to evolve male-type aggressive actions without altering courtship motor programs

In theory, presence or absence of just one gene can specify two sexes. However, most animal species use multiple genes for sex determination, even if one 'master' switch gene (such as *Sex-lethal* in *Drosophila*) may initiate the sex determination process (*Robinett et al., 2010*; *Williams and Carroll, 2009*). Our results suggest that one of the reasons behind this apparent genetic cooperativity may be that each gene controls specific aspects of sexually dimorphic traits, including behaviors (*Pereira et al., 2019*; *Xu et al., 2012*). We further speculate that separate genetic mechanisms for establishing sexually dimorphic actions and target sex-dependent action bias may be evolutionarily adaptive.

Courtship and aggression are likely under different types of selection pressures. Courtship behavior is executed by a male toward a female, who either rejects or accepts courtship (*Hall, 1994*). In *Ishii et al. (2020)*, we showed that *dsx* plays an important role in specifying the male courtship execution circuit. The *dsx* gene produces functional proteins in both sexes, and mediates specification of not only male-type circuits for courtship but also female-type circuits that make females receptive to such motor programs (*Deutsch et al., 2019*; *Zhou et al., 2014*). This dual function of *dsx* in males and females may ensure that the species-specificity of courtship rituals are preserved between both

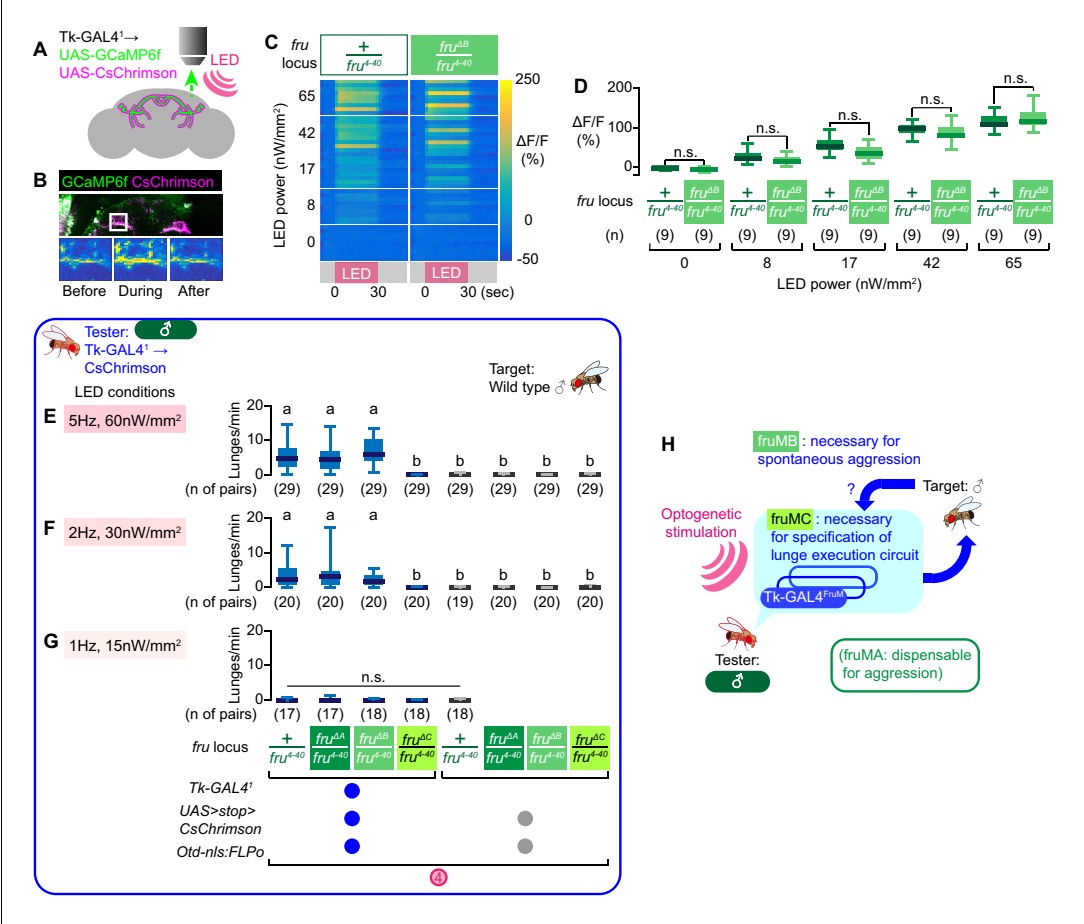

**Figure 6.** Tk-GAL4[FruM] neurons in fruMB mutants can promote male-type aggression. (A) Schematics of the functional imaging strategy. (B) Fluorescence of GCaMP6f (green) and CsChrimson:tdTomato (magenta), both of which are driven by *Tk-GAL4[1]* (top) and pseudocolored intensity of GCaMP6f fluorescence (within the area indicated by the white square in the top panel) before (bottom left), during (bottom middle), and after (bottom right) the LED stimulation, visualized by 2-photon microscopy. (C) Time course of pseudocolored fluorescence intensity (reference: right side) in *Tk-GAL4[1]; UAS-GCaMP6f; UAS-CsChrimson* male brains, with the *fru* locus of *+/fru[4-40]* (left) or *fru[ΔB]/fru[4-40]* (right). LED stimulation (pink at the bottom) was for 30 s at 5 Hz. LED powers are indicated on the left. (D) Boxplots of GCaMP6f fluorescence intensity changes in Tk-GAL4[FruM] neurons (data shown in C). Genotypes at the *fru* locus, number of brains examined, and LED powers are indicated below. (F–G) Boxplots of lunges by the tester flies under strong (E), medium (F), or weak (G) LED stimulation conditions (indicated in shaded pink boxes) during the time windows 4. Their genotypes and numbers are indicated below the plots. Lowercase letters denote significance groups (p<0.01, Kruskal-Wallis one-way ANOVA and post-hoc Dunn's multiple comparison test). (H): Schematic summary of the contributions of *fru* isoforms on male-type aggressive behavior.

The online version of this article includes the following figure supplement(s) for figure 6:

**Figure supplement 1.** Behaviors induced by optogenetic stimulation of male Tk-GAL4[FruM] neurons across various LED conditions.

sexes even during speciation, which must accompany co-evolution of both a male's courtship behavior and a female's choice process through sexual selection (*Majerus, 1986*). Interestingly, *dsx*-expressing neurons show high selectivity to courtship pulse song with behaviorally relevant inter-pulse interval frequencies both in males and females (*Deutsch et al., 2019*; *Zhou et al., 2015*).

In contrast, aggressive behavior is executed by sexually dimorphic motor programs often between two members of the same sex, most predominantly among males (*Chen et al., 2002*; *Nilsen et al., 2004*; *Smith and Price, 1973*). Consequently, aggressive actions can be under a different selection pressure from courtship behavior. Ample examples of male-specific weaponry organs across a variety of species support the hypothesis that sexually dimorphic traits that bring advantage specifically during inter-male competition can be fixed through evolution even if it is not directly under sexual selection (*Rico-Guevara and Hurme, 2018*). This hypothesis suggests that such traits, which can be behavioral as well as anatomical, may evolve independently of courtship behavior.

Indeed, some closely related Hawaiian *Drosophila* species evolved species-specific aggressive rituals even though action patterns of courtship behavior as well as morphology of antennae, an organ necessary for successful courtship, remain relatively similar (*Kaneshiro and Boake, 1987*; *Price and Boake, 1995*; *Spieth, 1981*). This example suggests that animals must be equipped with a genetic mechanism that controls motor programs for aggressive behavior at least partially separately from the motor programs for courtship behavior. *fru* is well situated to be a genetic agent for such a male-specific modification of aggressive behavior (*Davis et al., 2000*), because it seems dispensable for specifying a neural circuit that generates at least certain aspects of courtship action patterns (see *Ishii et al., 2020*). At the same time, *dsx* can ensure that males do not act aggressively toward a conspecific female. Female-type somatic characteristics, including sex-specific chemical cues (*Shirangi et al., 2009*), are specified by *dsx*. It is possible that dsxM prevents costly male aggression toward females by helping the brain process female-type sensory cues as an aggression-inhibiting signal (*Fernández et al., 2010*) in a species-specific manner.

The origin of sexual dimorphism in social behavior is a fundamental neuroscientific and ethological question. While circuit manipulations offer a powerful approach to dissect the neural mechanisms behind sexually dimorphic behaviors, comparative studies of the neural circuits controlling social behaviors in genetic gynandromorphs can serve as a complementary approach to address this question from both developmental, and possibly evolutionary, perspectives. Although sex-determining genetic pathways are diverse across animal species, studying *Drosophila* sexual dimorphism can lead to logical predictions about the genetic or neuronal mechanisms underlying sexual dimorphism of social behavior in mammals.

## Materials and methods

See *Supplementary file 1* for details of reagents used in this study.

### Fly strains

See *Table 1* for the complete genotypes of *Drosophila* strains used in each figure panel. *Tk-GAL4[1]* (RRID:BDSC_51975) and *Otd-nls:FLPo* (in attP40) were previously described *Asahina et al. (2014)*. *NP263*1 (*Yu et al., 2010*) is a gift from Daisuke Yamamoto (Tohoku University). *20XUAS > myr: TopHAT2 >CsChrimson:tdTomato* (in VK00022, VK00005 and attP2) (*Duistermars et al., 2018*; *Watanabe et al., 2017*), *20XUAS-IVS-Syn21-GCaMP6f* (codon-optimized)-p10 (in su(Hw)attP5), and *13XLexAop2-IVS-Syn21-GCaMP6f* (codon-optimized)-p10 (in su(Hw)attP5) were created by Barret Pfeiffer in the lab of Gerald Rubin (HHMI Janelia Research Campus) and kindly shared by David Anderson (California Institute of Technology). *fru[M]* (RRID:BDSC_66874), *fru[F]* (RRID:BDSC_66873) (*Demir and Dickson, 2005*), and *fru[FLP]* (RRID:BDSC_66870) (*Yu et al., 2010*) flies are gifts from Barry Dickson (HHMI Janelia Research Campus); *dsx[FLP]* (*Rezával et al., 2014*), *fru[ΔA]*, *fru[ΔB]* (*Neville et al., 2014*), and *fru[ΔC]* flies (*Billeter et al., 2006*) are gifts from Stephen Goodwin (University of Oxford); *fru[P1.LexA]* (RRID:BDSC_66698) (*Mellert et al., 2010*) is a gift from Bruce Baker (HHMI Janelia Research Campus); isogenic Canton-S, *fru[B1]*, and *fru[B2]* flies (*von Philipsborn et al., 2014*) are gifts from Anne von Philipsborn (University of Aarhus). *fru[4-40]* (RRID:BDSC_66692) was obtained from Bloomington *Drosophila* Resource Center in the University of Indiana. See *Ishii et al. (2020)* for the details of *R71G01-LexA*.

To create male flies that lack FruMA, FruMB, or FruMC, we followed approaches taken in previous publications (*Neville et al., 2014*; *von Philipsborn et al., 2014*) and created trans-heterozygotes of isoform-specific mutations and *fru[4-40]* or *fru[F]*. These two alleles do not transcribe fruM, while an isoform-specific mutation does not transcribe one of fruA, fruB, or fruC. Therefore, each trans-heterozygote is lacking a *fru* allele that can generate fruMA, fruMB, or fruMC transcripts, respectively.

### Immunohistochemistry

The following antibodies were used for immunohistochemistry with dilution ratios as indicated: rabbit anti-DsRed (1:1,000, Clontech # 632496, RRID:AB_10013483), mouse anti-BRP (1:100; Developmental Studies Hybridoma Bank nc82 (concentrated), RRID:AB_2314866), rabbit anti-FruM (1:10,000, a gift from Barry Dickson; *Stockinger et al., 2005*), guinea pig anti-FruM (1:100), rat anti-DsxM (1:100) (both gifts from Michael Perry, University of California, San Diego), rat anti-HA (1:100,

**Table 1.** Complete genotypes of *Drosophila* strains used in this study.

| FIGURE | PANEL | ABBREVIATED GENOTYPE | COMPLETE GENOTYPE ('Y' represents the Y chromosome) |
|---|---|---|---|
| *Figure 1* | C, D | *Tk-GAL4$^1$, UAS>stop> CsChrimson, Otd-nls:FLPo* | *w, Tk-GAL4$^1$/Y; Otd-nls:FLPo in attP40/+; 20XUAS > myr:TopHAT2 > CsChrimson: tdTomato in attP2, fru$^{4-40}$/+* |
| *Figure 1—figure supplement 1* | A-C | | |
| *Video 1* | | | |
| *Figure 1* | C, D | *Tk-GAL4$^1$, UAS > stop > CsChrimson* | *w, Tk-GAL4$^1$/Y; +/+; 20XUAS > myr: TopHAT2 > CsChrimson:tdTomato in attP2, fru$^{4-40}$/+* |
| *Figure 1* | C, D | *UAS > stop > CsChrimson, Otd-nls:FLPo* | *w/Y; Otd-nls:FLPo in attP40/+; 20XUAS > myr:TopHAT2 > CsChrimson: tdTomato in attP2, fru$^{4-40}$/+* |
| *Figure 1* | G, K, L, N-P | *Tk-GAL4$^1$, XY, fru locus: fru$^M$/fru$^{4-40}$* | *w, Tk-GAL4$^1$/Y; Otd-nls:FLPo in attP40/+; 20XUAS > myr:TopHAT2 > CsChrimson: tdTomato in attP2, fru$^{4-40}$/fru$^M$* |
| *Figure 1—figure supplement 2* | I | | |
| *Figure 1—figure supplement 3* | C | | |
| *Video 2, 3* | | | |
| *Figure 1* | H, K, M-P | *Tk-GAL4$^1$, XX, frulocus: fru$^M$/fru$^{4-40}$* | *w, Tk-GAL4$^1$/w, Tk-GAL4$^1$; Otd-nls:FLPo in attP40/+; 20XUAS > myr:TopHAT2 > CsChrimson:tdTomato in attP2, fru$^{4-40}$/fru$^M$* |
| *Figure 1—figure supplement 3* | C | | |
| *Video 3* | | | |
| *Figure 1* | I, K | *Tk-GAL4$^1$, XY, fru locus: fru$^F$/fru$^{4-40}$* | *w, Tk-GAL4$^1$/Y; Otd-nls:FLPo in attP40/+; 20XUAS > myr:TopHAT2 > CsChrimson: tdTomato in attP2, fru$^{4-40}$/fru$^F$* |
| *Figure 1* | J, K | *Tk-GAL4$^1$, XX, fru locus: fru$^F$/fru$^{4-40}$* | *w, Tk-GAL4$^1$/w, Tk-GAL4$^1$; Otd-nls:FLPo in attP40/+; 20XUAS > myr:TopHAT2 > CsChrimson:tdTomato in attP2, fru$^{4-40}$/fru$^F$* |
| *Figure 1* | K, N-P | *Tk-GAL4$^1$, XY, fru locus: +/+* | *w, Tk-GAL4$^1$/Y; Otd-nls:FLPo in attP40/+; 20XUAS > myr:TopHAT2 > CsChrimson: tdTomato in attP2/+* |
| *Figure 1—figure supplement 2* | C, D, H, J | | |
| *Figure 1—figure supplement 3* | C | | |
| *Figure 1* | K | *Tk-GAL4$^1$, XX, fru locus: +/+* | *w, Tk-GAL4$^1$/w, Tk-GAL4$^1$; Otd-nls:FLPo in attP40/+; 20XUAS > myr:TopHAT2 > CsChrimson:tdTomato in attP2/+* |
| *Figure 1—figure supplement 2* | K | | |
| *Figure 1* | K | *Tk-GAL4$^1$, XY, fru locus: +/fru$^{4-40}$* | *w, Tk-GAL4$^1$/Y; Otd-nls:FLPo in attP40/+; 20XUAS > myr:TopHAT2 > CsChrimson: tdTomato in attP2, fru$^{4-40}$/+* |
| *Figure 1—figure supplement 2* | L | | |
| *Figure 1* | K | *Tk-GAL4$^1$, XX, fru locus: +/fru$^{4-40}$* | *w, Tk-GAL4$^1$/w, Tk-GAL4$^1$; Otd-nls:FLPo in attP40/+; 20XUAS > myr:TopHAT2 > CsChrimson:tdTomato in attP2, fru$^{4-40}$/+* |
| *Figure 1—figure supplement 2* | M | | |
| *Figure 1—figure supplement 1* | D | *Tk-GAL4$^1$, UAS > stop > CsChrimson, Otd-nls:FLPo, fru locus: +/+* | *w, Tk-GAL4$^1$/Y; Otd-nls:FLPo in attP40/+; 20XUAS > myr:TopHAT2 > CsChrimson: tdTomato in attP2/+* |
| *Figure 1—figure supplement 2* | N | | |
| *Figure 1—figure supplement 1* | D | *Tk-GAL4$^1$, UAS > stop > CsChrimson, Otd-nls: FLPo, fru locus: +/fru$^{4-40}$* | *w, Tk-GAL4$^1$/Y; Otd-nls:FLPo in attP40/+; 20XUAS > myr:TopHAT2 > CsChrimson: tdTomato in attP2, fru$^{4-40}$/+* |
| *Figure 1—figure supplement 2* | N | | |

*Table 1 continued on next page*

*Table 1 continued*

| FIGURE | PANEL | ABBREVIATED GENOTYPE | COMPLETE GENOTYPE ('Y' represents the Y chromosome) |
|---|---|---|---|
| *Figure 1—figure supplement 1* | D | *Tk-GAL4$^1$, UAS > stop > CsChrimson, Otd-nls:FLPo, fru locus: fru$^M$/fru$^{4-40}$* | *w, Tk-GAL4$^1$/Y; Otd-nls:FLPo in attP40/+; 20XUAS > myr:TopHAT2 > CsChrimson: tdTomato in attP2, fru$^{4-40}$/fru$^M$* |
| *Figure 1—figure supplement 2* | N | | |
| *Figure 1—figure supplement 1* | D | *Tk-GAL4$^1$, UAS > stop > CsChrimson, fru locus: fru$^M$/fru$^{4-40}$* | *w, Tk-GAL4$^1$/Y; +/+; 20XUAS > myr:TopHAT2 > CsChrimson:tdTomato in attP2, fru$^{4-40}$/fru$^M$* |
| *Figure 1—figure supplement 1* | D | *UAS > stop > CsChrimson, Otd-nls: FLPo, fru locus: fru$^M$/fru$^{4-40}$* | *w/Y; Otd-nls:FLPo in attP40/+; 20XUAS > myr: TopHAT2 > CsChrimson:tdTomato in attP2, fru$^{4-40}$/fru$^M$* |
| *Figure 1—figure supplement 2* | F, G | Tk-GAL4$^1$ ∩ fru (genetic intersection) | *w, Tk-GAL4$^1$/Y; +/+; 20XUAS > myr:TopHAT2 > CsChrimson:tdTomato in attP2/fru$^{FLP}$* (2 samples for averaged image in F) |
| *Figure 1—figure supplement 2*  Video 2 | G | Tk-GAL4$^1$ ∩ fru (genetic intersection) | *w, Tk-GAL4$^1$/Y; +/+; 20XUAS>myr:TopHAT2 > CsChrimson:tdTomato in VK00005/fru$^{FLP}$* (three samples) *w, Tk-GAL4$^1$/Y; 8XLexAop2-FLPL in attP40/+; 20XUAS > myr:TopHAT2 > CsChrimson: tdTomato in attP2/fru$^{P1.LexA}$* (2 samples) |
| *Figure 1—figure supplement 3* | A, C | Tk-GAL4$^1$, XY, fru locus: fru$^F$/+ | *w, Tk-GAL4$^1$/Y; Otd-nls:FLPo in attP40/+; 20XUAS > myr:TopHAT2 > CsChrimson: tdTomato in attP2, fru$^F$/+* |
| *Figure 1—figure supplement 3* | B, C | Tk-GAL4$^1$, XX, fru locus: fru$^M$/+ | *w, Tk-GAL4$^1$/w, Tk-GAL4$^1$; Otd-nls:FLPo in attP40/+; 20XUAS > myr:TopHAT2 > CsChrimson:tdTomato in attP2/fru$^M$* |
| *Figure 1—figure supplement 3* | E | *Tk-GAL4$^1$, UAS > stop > CsChrimson (attP2), Otd-nls:FLPo, fru locus: +/+* | *w, Tk-GAL4$^1$/Y; Otd-nls:FLPo in attP40/+; 20XUAS > myr:TopHAT2 > CsChrimson: tdTomato in attP2/+* |
| *Figure 1—figure supplement 3* | E | *Tk-GAL4$^1$, UAS > stop > CsChrimson (attP2), fru locus: fru$^{FLP}$/+* | *w, Tk-GAL4$^1$/Y; +/+; 20XUAS > myr:TopHAT2 > CsChrimson:tdTomato in attP2/fru$^{FLP}$* |
| *Figure 1—figure supplement 3* | E | *Tk-GAL4$^1$, UAS > stop > CsChrimson (VK00005), fru locus: fru$^{FLP}$/+* | *w, Tk-GAL4$^1$/Y; +/+; 20XUAS > myr:TopHAT2 > CsChrimson:tdTomato in VK00005, fru$^{FLP}$/+* |
| *Figure 1—figure supplement 3* | E | *Tk-GAL4$^1$, UAS>stop>CsChrimson (VK00005), fru locus:fru$^{FLP}$/fru$^{4-40}$* | *w, Tk-GAL4$^1$/Y; +/+; 20XUAS > myr:TopHAT2 > CsChrimson:tdTomato in VK00005, fru$^{FLP}$/fru$^{4-40}$* |
| *Figure 2* | A,-C | fruM, *Tk-GAL4$^1$, UAS > stop > CsChrimson, Otd-nls:FLPo* | *w, Tk-GAL4$^1$/w, Tk-GAL4$^1$; Otd-nls:FLPo in attP40/+; 20XUAS > myr:TopHAT2 > CsChrimson:tdTomato in attP2, fru$^{4-40}$/fru$^M$* |
| *Figure 2—figure supplement 2* | A, B | | |
| *Figure 2—figure supplement 3*  Video 4, 5 | A, B, E | | |
| *Figure 2* | A-C | fruM, *Tk-GAL4$^1$, UAS > stop > CsChrimson* | *w, Tk-GAL4$^1$/w, Tk-GAL4$^1$; +/+; 20XUAS > myr:TopHAT2 > CsChrimson: tdTomato in attP2, fru$^{4-40}$/fru$^M$* |
| *Figure 2—figure supplement 3* | A, B | | |
| *Figure 2* | A-C | fruM, *UAS > stop > CsChrimson, Otd-nls:FLPo* | *w/w; Otd-nls:FLPo in attP40/+; 20XUAS > myr:TopHAT2 > CsChrimson: tdTomato in attP2, fru$^{4-40}$/fru$^M$* |
| *Figure 2—figure supplement 3* | A, B | | |
| *Figure 2* | C | *Tk-GAL4$^1$, UAS > stop > CsChrimson, Otd-nls:FLPo* | *w, Tk-GAL4$^1$/Y; Otd-nls:FLPo in attP40/+; 20XUAS > myr: TopHAT2 > CsChrimson:tdTomato in attP2, fru$^{4-40}$/+* |
| *Figure 2—figure supplement 3* | D | | |
| *Figure 2—figure supplement 1* | A, B | Wild type (tester) | +/+; +/+; +/+ (Canton-S) |
| | | (target) | *w/Y; +/20XUAS > myr:TopHAT2 > CsChrimson: tdTomato in VK00022; dsx$^{FLP}$, fru$^{4-40}$/fru$^M$* |
| | | fruF (target) | *w/Y; +/20XUAS > myr:TopHAT2 > CsChrimson: tdTomato in VK00022; dsx$^{FLP}$, fru$^{4-40}$/fru$^F$* |
| *Figure 2—figure supplement 3* | C, F | fruM, *Tk-GAL4$^1$ (hetero), UAS > stop > CsChrimson, Otd-nls:FLPo* | *w, Tk-GAL4$^1$/w; Otd-nls:FLPo in attP40/+; 20XUAS > myr:TopHAT2 > CsChrimson: tdTomato in attP2, fru$^{4-40}$/fru$^M$* |

*Table 1 continued*

| FIGURE | PANEL | ABBREVIATED GENOTYPE | COMPLETE GENOTYPE ('Y' represents the Y chromosome) |
|---|---|---|---|
| *Figure 2—figure supplement 3* | C | fruM, *Tk-GAL4[1]* (hetero), *UAS > stop > CsChrimson* | *w, Tk-GAL4[1]/w; +/+; 20XUAS > myr:TopHAT2 > CsChrimson:tdTomato in attP2, fru[4-40]/fru[M]* |
| *Figure 3* | B, C | *NP2631, UAS > stop > CsChrimson, dsx[FLP]* | *w/w; NP2631/20XUAS > myr:TopHAT2 > CsChrimson:tdTomato in VK00022; dsx[FLP], fru[4-40]/fru[F]* |
| *Figure 3—figure supplement 1* | A, B | | |
| *Video 6* | | | |
| *Figure 3* | B, C | *NP2631, UAS > stop > CsChrimson* | *w/w; NP2631/20XUAS > myr:TopHAT2 > CsChrimson:tdTomato in VK00022; fru[4-40]/fru[F]* |
| *Figure 3* | B, C | *UAS > stop > CsChrimson, dsx[FLP]* | *w/w; +/20XUAS > myr:TopHAT2 > CsChrimson: tdTomato in VK00022; dsx[FLP], fru[4-40]/fru[F]* |
| *Figure 3* | D, E | fruM, *NP2631, UAS > stop > CsChrimson, dsx[FLP]* | *w/w; NP2631/20XUAS > myr:TopHAT2 > CsChrimson:tdTomato in VK00022; dsx[FLP], fru[4-40]/fru[M]* |
| *Figure 3* | D, E | fruM, *NP2631, UAS > stop > CsChrimson* | *w/w; NP2631/20XUAS > myr:TopHAT2 > CsChrimson:tdTomato in VK00022; fru[4-40]/fru[M]* |
| *Figure 3* | D, E | fruM, *UAS > stop > CsChrimson, dsx[FLP]* | *w/w; +/20XUAS > myr:TopHAT2 > CsChrimson: tdTomato in VK00022; dsx[FLP], fru[4-40]/fru[M]* |
| *Figure 3—figure supplement 1* | C | *NP2631, UAS > stop > CsChrimson, dsx[FLP] fru locus: +/+* | *w/w; NP2631/20XUAS > myr:TopHAT2 > CsChrimson:tdTomato in VK00022; dsx[FLP]/+* |
| *Figure 3—figure supplement 1* | C | *NP2631, UAS > stop > CsChrimson, dsx[FLP] fru locus: +/fru[4-40]* | *w/w; NP2631/20XUAS > myr:TopHAT2 > CsChrimson:tdTomato in VK00022; dsx[FLP], fru[4-40]/+* |
| *Figure 3—figure supplement 1* | C | *NP2631, UAS > stop > CsChrimson, dsx[FLP] fru locus: fru[F]/fru[4-40]* | *w/w; NP2631/20XUAS > myr:TopHAT2 > CsChrimson:tdTomato in VK00022; dsx[FLP], fru[4-40]/fru[F]* |
| *Figure 3—figure supplement 1* | D, E | *NP2631 ∩ dsx[FLP], R71G01-LexA* | *w/w; NP2631/13XLexAop2-IVS-GCaMP6f-p10 in su(Hw)attP5; 20XUAS > myr:TopHAT2 > CsChrimson:tdTomato in VK00005, dsx[FLP]/R71G01-LexA in attP2* |
| *Figure 4* | B | *fru locus: +/fru[4-40]* | *+/Y; +/+; +/fru[4-40]* (F1 hybrid of +; +; fru[4-40]/TM6B and isogenic Canton-S) |
| *Figure 4—figure supplement 1* | A-E | | |
| *Figure 4* | B | *fru locus: fru[ΔA]/fru[4-40]* | *+/Y; +/+; fru[ΔA]/fru[4-40]* |
| *Figure 4—figure supplement 1* | A, B | | |
| *Figure 4* | B | *fru locus: fru[ΔB]/fru[4-40]* | *+/Y; +/+; fru[ΔB]/fru[4-40]* |
| *Figure 4—figure supplement 1* | A, B, D, E | | |
| *Figure 4* | B | *fru locus: fru[ΔC]/fru[4-40]* | *+/Y; +/+; fru[ΔC]/fru[4-40]* |
| *Figure 4—figure supplement 1* | A, B | | |
| *Figure 4* | B | *fru locus: +/+* | *+/Y; +/+; +/+* (F1 hybrid of Canton-S and isogenic Canton-S) |
| *Figure 4—figure supplement 1* | A-E | | |
| *Figure 4* | B | *fru locus: fru[ΔA]/+* | *+/Y; +/+; fru[ΔA]/+* |
| *Figure 4—figure supplement 1* | A, B | | |
| *Figure 4* | B | *fru locus: fru[ΔB]/+* | *+/Y; +/+; fru[ΔB]/+* |
| *Figure 4—figure supplement 1* | A, B, D, E | | |
| *Figure 4* | B | *fru locus: fru[ΔC]/+* | *+/Y; +/+; fru[ΔC]/+* |
| *Figure 4—figure supplement 1* | A, B | | |
| *Figure 4* | C | *fru locus: +/fru[F]* | *+/Y; +/+; +/fru[F]* |

*Table 1 continued on next page*

*Table 1 continued*

| FIGURE | PANEL | ABBREVIATED GENOTYPE | COMPLETE GENOTYPE ('Y' represents the Y chromosome) |
|---|---|---|---|
| *Figure 4* | C | *fru* locus: $fru^{\Delta B}/fru^{F}$ | +/Y; +/+; $fru^{\Delta B}/fru^{F}$ |
| *Figure 4* | C | *fru* locus: $+/fru^{M}$ | +/Y; +/+; $+/fru^{M}$ |
| *Figure 4* | C | *fru* locus: $fru^{\Delta B}/fru^{M}$ | +/Y; +/+; $fru^{\Delta B}/fru^{M}$ |
| *Figure 4—figure supplement 1* | C | *fru* locus: $fru^{B1}/fru^{4-40}$ | +/Y; +/+; $fru^{B1}/fru^{4-40}$ |
| *Figure 4—figure supplement 1* | C | *fru* locus: $fru^{B2}/fru^{4-40}$ | +/Y; +/+; $fru^{B2}/fru^{4-40}$ |
| *Figure 4—figure supplement 1* | C | *fru* locus: $fru^{B1}/+$ | +/Y; +/+; $fru^{B1}/+$ |
| *Figure 4—figure supplement 1* | C | *fru* locus: $fru^{B2}/+$ | +/Y; +/+; $fru^{B2}/+$ |
| Figure 5 / Video 7 | A, E, F, I-K | Tk-GAL4$^1$, XY, *fru* locus: $+/fru^{4-40}$ | w, Tk-GAL4$^1$/Y; Otd-nls:FLPo in attP40/+; 20XUAS > myr:TopHAT2 > CsChrimson: tdTomato in attP2, $fru^{4-40}/+$ |
| Figure 5 | B, E, G, I-K | Tk-GAL4$^1$, XY, *fru* locus: $fru^{\Delta A}/fru^{4-40}$ | w, Tk-GAL4$^1$/Y; Otd-nls:FLPo in attP40/+; 20XUAS > myr:TopHAT2 > CsChrimson: tdTomato in attP2, $fru^{4-40}/fru^{\Delta A}$ |
| Figure 5 | C, E, H-K | Tk-GAL4$^1$, XY, *fru* locus: $fru^{\Delta B}/fru^{4-40}$ | w, Tk-GAL4$^1$/Y; Otd-nls:FLPo in attP40/+; 20XUAS > myr:TopHAT2 > CsChrimson: tdTomato in attP2, $fru^{4-40}/fru^{\Delta B}$ |
| Figure 5 | D, E | Tk-GAL4$^1$, XY, *fru* locus: $fru^{\Delta C}/fru^{4-40}$ | w, Tk-GAL4$^1$/Y; Otd-nls:FLPo in attP40/+; 20XUAS > myr:TopHAT2 > CsChrimson: tdTomato in attP2, $fru^{4-40}/fru^{\Delta C}$ |
| Figure 5 / Figure 5—figure supplement 1 | E / A | Tk-GAL4$^1$, XY, *fru* locus: $+/+$ | w, Tk-GAL4$^1$/Y; Otd-nls:FLPo in attP40/+; 20XUAS > myr:TopHAT2 > CsChrimson: tdTomato in attP2/+ |
| Figure 5 / Figure 5—figure supplement 1 | E / B | Tk-GAL4$^1$, XY, *fru* locus: $fru^{\Delta A}/+$ | w, Tk-GAL4$^1$/Y; Otd-nls:FLPo in attP40/+; 20XUAS > myr:TopHAT2 > CsChrimson: tdTomato in attP2/$fru^{\Delta A}$ |
| Figure 5 / Figure 5—figure supplement 1 | E / C | Tk-GAL4$^1$, XY, *fru* locus: $fru^{\Delta B}/+$ | w, Tk-GAL4$^1$/Y; Otd-nls:FLPo in attP40/+; 20XUAS > myr:TopHAT2 > CsChrimson: tdTomato in attP2/$fru^{\Delta B}$ |
| Figure 5 / Figure 5—figure supplement 1 | E / D | Tk-GAL4$^1$, XY, *fru* locus: $fru^{\Delta C}/+$ | w, Tk-GAL4$^1$/Y; Otd-nls:FLPo in attP40/+; 20XUAS > myr:TopHAT2 > CsChrimson: tdTomato in attP2/$fru^{\Delta C}$ |
| Figure 6 | B-D | Tk-GAL4$^1$, UAS-GCaMP6f, UAS-CsChrimson, *fru* locus: $+/fru^{4-40}$ | w, Tk-GAL4$^1$/Y; Otd-nls:FLPo in attP40/20XUAS-IVS-Syn21-GCaMP6f-p10 in su(Hw)attP5; 20XUAS > myr:TopHAT2 > CsChrimson:tdTomato in attP2, $fru^{4-40}/+$ |
| Figure 6 | C, D | Tk-GAL4$^1$, UAS-GCaMP6f, UAS-CsChrimson, *fru* locus: $fru^{\Delta B}/fru^{4-40}$ | w, Tk-GAL4$^1$/Y; Otd-nls:FLPo in attP40/20XUAS-IVS-Syn21-GCaMP6f-p10 in su(Hw)attP5; 20XUAS > myr:TopHAT2 > CsChrimson:tdTomato in attP2, $fru^{4-40}/fru^{\Delta B}$ |
| Figure 6 / Figure 6—figure supplement 1 | E-G / A-C | *Tk-GAL4$^1$, UAS > stop > CsChrimson, Otd-nls:FLPo,* *fru* locus: $+/fru^{4-40}$ | w, Tk-GAL4$^1$/Y; Otd-nls:FLPo in attP40/+; 20XUAS > myr:TopHAT2 > CsChrimson:tdTomato in attP2, $fru^{4-40}/+$ |
| Figure 6 / Figure 6—figure supplement 1 | E-G / A-C | *Tk-GAL4$^1$, UAS > stop > CsChrimson, Otd-nls:FLPo,* *fru* locus: $fru^{\Delta A}/fru^{4-40}$ | w, Tk-GAL4$^1$/Y; Otd-nls:FLPo in attP40/+; 20XUAS > myr: TopHAT2 > CsChrimson:tdTomato in attP2, $fru^{4-40}/fru^{\Delta A}$ |
| Figure 6 / Figure 6—figure supplement 1 | E-G / A-C | *Tk-GAL4$^1$, UAS > stop > CsChrimson, Otd-nls:FLPo,* *fru* locus: $fru^{\Delta B}/fru^{4-40}$ | w, Tk-GAL4$^1$/Y; Otd-nls:FLPo in attP40/+; 20XUAS > myr: TopHAT2 > CsChrimson:tdTomato in attP2, $fru^{4-40}/fru^{\Delta B}$ |

*Table 1 continued on next page*

*Table 1 continued*

| FIGURE | PANEL | ABBREVIATED GENOTYPE | COMPLETE GENOTYPE ('Y' represents the Y chromosome) |
|---|---|---|---|
| *Figure 6* | E-G | *Tk-GAL4[1], UAS > stop > CsChrimson, Otd-nls: FLPo,* *fru* locus: *fru$^{\Delta C}$/fru$^{4-40}$* | *w, Tk-GAL4[1]/Y; Otd-nls:FLPo in attP40/+; 20XUAS > myr: TopHAT2 > CsChrimson:tdTomato in attP2, fru$^{4-40}$/fru$^{\Delta C}$* |
| *Figure 6—figure supplement 1* | A-C | | |
| *Figure 6* | E-G | *UAS > stop > CsChrimson, Otd-nls:FLPo, fru* locus: *+/fru$^{4-40}$* | *w/Y; Otd-nls:FLPo in attP40/+; 20XUAS > myr:TopHAT2 > CsChrimson: tdTomato in attP2, fru$^{4-40}$/+* |
| *Figure 6* | E-G | *UAS > stop > CsChrimson, Otd-nls:FLPo, fru* locus: *fru$^{\Delta A}$/fru$^{4-40}$* | *w/Y; Otd-nls:FLPo in attP40/+; 20XUAS > myr:TopHAT2 > CsChrimson: tdTomato in attP2, fru$^{4-40}$/fru$^{\Delta A}$* |
| *Figure 6* | E-G | *UAS > stop > CsChrimson, Otd-nls:FLPo, fru* locus: *fru$^{\Delta B}$/fru$^{4-40}$* | *w/Y; Otd-nls:FLPo in attP40/+; 20XUAS > myr:TopHAT2 > CsChrimson: tdTomato in attP2, fru$^{4-40}$/fru$^{\Delta B}$* |
| *Figure 6* | E-G | *UAS > stop > CsChrimson, Otd-nls:FLPo, fru* locus: *fru$^{\Delta B}$/fru$^{4-40}$* | *w/Y; Otd-nls:FLPo in attP40/+; 20XUAS > myr:TopHAT2 > CsChrimson: tdTomato in attP2, fru$^{4-40}$/fru$^{\Delta C}$* |

Roche Cat# 11867423001, RRID:AB_390918), goat anti-rat Alexa 488 (1:100, ThermoFisher Scientific Cat# A11006, RRID:AB_2534074), goat anti-rabbit Alexa 488 (1:100, ThermoFisher Scientific, Cat# A11034, RRID:AB_2576217), goat anti-rat Alexa 488 (1:100, ThermoFisher Scientific Cat# A11006), goat anti-rabbit Alexa 568 (1:100; ThermoFisher Scientific Cat# A11036, RRID:AB_10563566), goat anti-mouse Alexa 633 (1:100; ThermoFisher Scientific Cat# A21052, RRID:AB_2535719), goat anti-guinea pig Alexa 633 (1:100, ThermoFisher Scientific Cat# A21105, RRID:AB_2535757). Immunohistochemistry of the fly brains followed the protocol described in *Van Vactor et al. (1991)*. Briefly, the fly brains are dissected in 1XPBS and fixed in 1XPBS with 2% formaldehyde and 75 mM L-lysine for 75–90 min at room temperature. The brains were then washed in PBST (1XPBS, 0.3% TritonX-100) and were incubated in the blocking solution (10% heat-inactivated normal goad serum, 1XPBS, 0.3% TritonX-100) for 30 min. Primary antibodies were diluted in the blocking solution and were applied to samples, which were then incubated at 4°C for 2 days. The brains were then washed in PBST and then incubated in the blocking solution for 30 min. Secondary antibodies were diluted in the blocking solution and were applied to the samples, which were then incubated at 4°C overnight. The brains were then washed in PBST, and then either incubated in 1XPBS, 50% glycerol for 2 hr at room temperature before mounted in Vectashield (Vector Laboratories, Cat# H-1000) onto a slide glass, or incubated in FocusClear (CelExplorer Labs, Taiwan, Cat# FC-101) medium for 2 hr at room temperature before being mounted in MountClear (CelExplorer Labs, Taiwan, Cat# MC-301) medium. A small well was made by cutting vinyl tape fixed on a slide glass, and one brain was transferred to each well before a cover slip (#1.5) was placed on the well and was sealed with nail polish. All reactions were carried out in a well of 6 × 10 microwell mini tray (ThermoFisher Scientific Cat# 439225).

For simultaneous detection of DsxM and FruM (*Figure 1—figure supplement 2C,D*), brains were fixed in 1XPBS with 4% formaldehyde for 15 min at room temperature. Brains were then transferred to a 1.5 mL microtube, in which the remaining steps were carried out.

Z-stack images were acquired by FV-1000 confocal microscopy (Olympus America) except samples for *Figure 3—figure supplement 1D, E*, which were acquired by a Zeiss 710 confocal microscopy (Carl Zeiss Microscopy) at the Salk Institute Biophotonics Core, and were processed in Fiji software (*Schindelin et al., 2012*) (RRID:SCR_002285; https://fiji.sc/). The despeckle function was applied before a z-projection image was generated using maximum intensity projection. Minimum or maximum intensity thresholds were adjusted for enhanced clarity. Source image files used in all figures can be found in *Source data 1*.

## Segmentation, registration, and analysis of immunohistochemical samples

All data points for anatomical quantifications used in all figures, as well as all statistical results with exact p values, can be found in *Source data 1*.

Parametric tests were applied as indicated in figure legends to compare cell body numbers among different genotypes. All data points have biological replicates of at least 8, which has sufficient power to detect mean changes in cell body number of larger than 1.75 when assuming a mean

cell body number of 5 and a standard deviation of 1.5 (which is reasonable based on our data). As indicated in figures, genotype-dependent changes in cell body number had a larger effect size than 2 in all cases.

For segmentation of Tk-GAL4$^{FruM}$ neurons, we first visualized each z-stack in a 3D space using a rendering software FluoRender (*Wan et al., 2009*) (RRID:SCR_014303; https://github.com/SCIInstitute/fluorender). We then chose the channel corresponding to Alexa 568 (which labels CsChrimson: tdTomato immunohistochemistry signal), and traced neural processes emanating from the cell bodies of Tk-GAL4$^{FruM}$ neurons, which could be identified unambiguously at the lateral side of the brain. The neural processes were then segmented using the Paint Brush function (*Wan et al., 2017*), and saved as a new. nrrd file.

For registration of brains, we split each channel of a z-stack file and resaved each as an individual. nrrd files in Fiji. We added the newly created. nrrd file that represent Tk-GAL4$^{FruM}$ neurons (see above) into the same folder, and used the Fiji plugin for Computational Morphometry Toolkit (CMTK) (RRID:SCR_002234; https://www.nitrc.org/projects/cmtk; *Rohlfing and Maurer, 2003*) to register the brain image stacks to the template brain as described in *Jefferis et al. (2007)* (https://github.com/jefferis/fiji-cmtk-gui). Briefly, for each brain, the image of neuropil visualized by anti-BRP antibody was used to transform the z-stack to the template brain, and the same transformation was subsequently applied to additional channels. Registration was performed by using the same parameters implemented in '*Cachero et al. (2010)*' (exploration = 26, coarsest = 8, grid spacing = 80, refine = 4, accuracy = 0.4) in CMTK plugin on Fiji. We used JRC2018 INTERSEX (*Bogovic et al., 2018*) as our template, since we experienced more robust registration result than with the FCWB template brain (*Costa et al., 2016*) (data not shown).

After the registration of z-stacks, we calculated the average of the images in a hemibrain for each genotype. We did this by horizontally flipping each transformed z-stack images, and calculating the average signal intensity in each voxel. The resulting averaged images are bilaterally symmetrical. Therefore, only the left hemisphere is shown for z-projection images.

To calculate the volumes of specific neuronal structures within Tk-GAL4$^{FruM}$ neurons, we further segmented the target structures using the Paint Brush function of FluoRender, and calculated the volume of each structure using the Volume Size function. Statistical analyses were carried out using MATLAB (The Mathworks, Inc, RRID:SCR_001622). The Kruskal-Wallis test ('kruskalwallis') was used to evaluate whether a volume of the given structure was significantly different among different genotypes. When the p-value was below 0.05, the post-hoc the Mann-Whitney U-test ('ranksum') was used to detect significant differences between testing and control genotypes. In both cases, Bonferroni correction was applied to p values. Non-parametric tests were applied for volume data since we could not necessarily assume the normal distribution of this data type.

## Social behavior analysis

### Subject preparation

Flies were collected on the day of eclosion into vials containing standard cornmeal-based food, and were kept either as a group of up to 16 flies per vial, or singly at 25 °C with 60% relative humidity, and a 9AM:9PM light:dark cycle. For optogenetic experiments, the tester flies were reared on food containing 0.2 mM all-*trans* retinal (MilliporeSigma, Cat#R2500, 20 mM stock solution prepared in 95% ethanol), and vials were covered with aluminum foil to shield light. Every 3 days, flies were transferred to vials containing fresh food. Tester flies were aged for 5–7 days except tester flies used in *Figure 3B–E* and *Figure 3—figure supplement 1A–C*, which were aged for 14–16 days to ensure consistent labeling of targeted neurons (data not shown).

Male target flies were group-reared Canton-S (originally from the lab of Martin Heisenberg, University of Würzberg) virgin males except those used in *Figure 4B–C* and *Figure 4—figure supplement 1C*, which were males of an isogenic Canton-S (a gift from Anne von Philipsborn, Aarhus University). They were used as targets in these experiments because we observed that our standard Canton-S target males were sometimes aggressive and dominant towards the tester males even after being reared in groups (data not shown), which we were concerned could obscure the innate level of aggression in the tester male flies. These isogenic males performed few lunges toward the tester male flies (data not shown). To prepare mated wild-type target females, 5 Canton-S males were introduced into vials with 10 virgin females at 4 days old, and were reared for 2 more days to

let them mate. At 3 days old, both male and female target flies were briefly anesthetized with $CO_2$, and the tip of either one of their wings were clipped by a razor to create a 'mark'. This clipping treatment did not reduce the amount of each behavior (lunge, wing extension, and headbutt) detected under our experimental settings (see *Ishii et al., 2020* for details).

## Behavioral assays

All behavior assays were conducted in the evening between Zeitberger time (ZT)7 and ZT12 (from 4 to 9PM) at 22–25℃. Social behavior assays were performed in a '12-well' acrylic chamber (*Asahina et al., 2014*) with food substrate (apple juice (Minute Maid) supplemented with 2.25% w/v agarose and 2.5% w/v sucrose; *Hoyer et al., 2008*) covering the entire floor of arena. The wall was coated with Insec-a-Slip (Bioquip Products, Inc, Cat# 2871C) and the ceiling was coated with Surfasil Siliconizing Fluid (ThermoFisher Scientific, Cat# TS-42800), both to prevent flies from climbing, as described previously (*Asahina et al., 2014*; *Hoyer et al., 2008*). The arenas were lit by LED back-lights, which were controlled by a custom-built switch box. For optogenetic experiments, 850 nm infrared LED backlights (Sobel Imaging Systems, CA, Cat# SOBL-150 × 100–850) were used, whereas white backlights (Edmunud Optics, NJ, Cat# 83873) were used for non-optogenetic experiments (*Figure 4A,B*, *Figure 4—figure supplement 1A–C*). Flies were introduced into the chamber by gentle aspiration, and were allowed to acclimate for 5 min before recording started.

Recording was done by USB3 digital cameras (Point Grey Flea3 USB3.0, FLIR Inc, Cat# FL3-U3-13Y3M-C) controlled by the BIAS acquisition software (IORodeo, CA; https://bitbucket.org/iorodeo/bias). The camera was equipped with a machine vision lens (Fujinon, Cat# HF35HA1B), and an infra-red longpass filter (Midwest Optical Systems, Cat# LP780-25.5) when the infrared light sources were used. Movies were taken at 60 frames per second in the AVI format, either for 10 min in the optoge-netic experiments or for 30 min for non-optogenetic experiments. Flies were discarded after each experiment. The food substrates were changed to a new one after 2 recordings for 30 min movies, or after 3 recordings for 10 min movies.

The setup for optogenetic experiments was assembled as described previously (*Inagaki et al., 2014*). Briefly, the red light (655 nm) LEDs were controlled via an Arduino Uno board (Arduino, Italy) using a custom program. As illustrated in *Figure 1B*, the stimulation paradigm (10 min in total) con-sists of 1 min pre-stimulation (time window '1' in *Figure 1B*), three blocks of 1 min stimulation at an indicated frequency (time window '2', 3 min in total) each followed by 2 min inter-stimulus intervals (ISIs, time window '3', 6 min in total). The recording and LED control were manually started simultaneously.

## Quantification of social behavior data

All behavioral data points used in all figures, as well as all statistical results with exact p values, can be found in *Source data 1*.

Acquired movies were first processed by the FlyTracker program (*Asahina et al., 2014*; http://www.vision.caltech.edu/Tools/FlyTracker/), which runs on MATLAB and creates output files which JAABA uses for behavioral classifications. See *Ishii et al. (2020)* for the creation, validation, and. jaab source files of JAABA-based behavioral classifiers used in this study, as well as the definition of 'time orienting'. The regions of interest were defined as circles corresponding to the chamber of each arena. The identities of tester and target flies were manually confirmed. The fly pair was removed from further analysis when (1) one of the two flies was killed during introduction to the chamber, (2) the wings of either fly was stuck at the extended position, or (3) the discrimination of the two flies was impossible due to wing damage of a tester fly. The amount of behavior is the num-ber of bouts for lunges and headbutts, and the total duration (seconds) for wing extensions. These amounts were binned per minute for quantification. Extremely short bouts detected by a classifier were almost always false positives, and were eliminated from quantification. For lunges and head-butts, events with duration of less than 50 milliseconds were discarded. For wing extensions, events with a duration of less than 100 milliseconds were discarded. The post-processing of data was done using custom MATLAB codes which are available on Github (https://github.com/wohlmp/Ishii_Wohl_DeSouza_Asahina_2019; *Wohl, 2019*; copy archived at https://github.com/elifesciences-publica-tions/Ishii_Wohl_DeSouza_Asahina_2019).

The frame in which the infrared indicator LED turned on during the first LED stimulation period was used to align frames of movies. Statistical analyses were carried out using MATLAB. After behavior within each time window were calculated (see *Figure 1B*), the Kruskal-Wallis test ('kruskalwallis') was used to evaluate whether a given behavior was significantly different among different illumination periods (periods 1, 2, and 3; *Figure 1B*) or among different genotypes. When the p-value was below 0.05, the post-hoc Mann-Whitney signed rank test ('signrank') was used to detect significant differences between illumination period, and the Mann-Whitney U-test ('ranksum') was used to detect significant differences between testing and control genotypes. In both cases, Bonferroni correction was applied to p values. When the uncorrected p value was less than 0.05 but the corrected value did not pass the significant level, the uncorrected value was shown on panels with parenthesis. For *Figure 6E–G*, Dunn's multiple comparison test was applied to detect significant groups using a custom code ('dunn.m') (*Cardillo, 2006*; http://www.mathworks.com/matlabcentral/fileexchange/12827).

While it is not feasible to perform power analyses for non-parametric datasets, we performed a bootstrapped power analysis with a sample dataset from 171 single-housed wild-type male fly pairs (which is not included in this manuscript). We asked how many pairs of flies (biological replicates) are necessary to achieve a power of 0.8 when attempting to detect a 50% difference in the number of lunges and found that 22 pairs are sufficient. Thirty pairs of flies would increase the power to 0.9. We tested at least 24 pairs per genotype for all social behavior assays, and many groups contain more than 30 pairs. While the data distribution appears different among different genotypes and behaviors, we are confident that the number of biological replicates used in this study gives us sufficient power to support our conclusions.

Generally, male testers seldom performed lunges toward female targets. Also, noticeably less lunges were detected in fruM females. Behavioral data from these conditions often result in zero-inflated datasets. In such cases, a few false positive incidents can impact the result of statistical tests. To avoid this pitfall, classifier results of lunges from male testers toward female targets and from fruM females were manually validated, and obvious false positives (caused by tracking errors, when a fly was near or on the wall, or when a fly suddenly jumped) were eliminated before statistical tests were applied.

## Activity and sleep analysis

Male flies were collected and singly reared for 6 days as was described in 'Social behavior analysis'. Experiments were carried out for 3 days at 24°C. Activity and sleep data were acquired using individual *Drosophila* activity monitors (TriKinetics Inc) under 12 hr:12 hr light:dark cycles, as was described previously (*Wu et al., 2014*). Counts of beam crosses between Zeitgerber (ZT)8 and ZT12 on day 2 and day 3 were binned per minute (note that ZT8-12 corresponds to the time period when we performed social behavioral assays). Sleep was defined as at least 5 min of inactivity (no beam cross). The amount of activity and sleep were quantified using custom MATLAB scripts provided by William Joiner (University of California, San Diego).

## Functional calcium imaging

Male flies were kept in all-*trans* retinal food after eclosion and aged 6 days. On the day of the experiment, flies were briefly anesthetized on ice and mounted on a custom chamber using ultraviolet curing adhesive (Norland Optical Adhesive 63, Norland Products, Inc) to secure the head and thorax to a tin foil base. The proboscis was also dabbed with glue to prevent its extension from altering the position of the brain. The head cuticle was removed with sharp forceps in room temperature *Drosophila* adult hemolymph-like saline (*Wang et al., 2003*). After cuticle removal, the recording solution was refreshed. Optogenetic stimulation was applied with an external fiber-coupled LED of 625 nm (Thorlabs Inc, Cat# M625F2) controlled by a programmable LED driver (ThorLabs, Cat# DC2200). The end of the LED fiber (Thorlabs, Cat# M28L01) was placed 5 mm from the brain. LED illumination of 5 Hz, 10-millisecond pulses for a 30 s duration was applied for optogenetic activation of CsChrimson. LED power was varied so that each fly received stimulations at 0, 0.02, 0.04, 0.1, and 0.2 mA. The energy from the LED that the neurons receive was estimated from the measurement of the LED power using a photodiode power sensor (Thorlabs, Cat# S130C) 5 mm away from the end

of the LED fiber. The energy was read using a digital optical power/energy meter (Thorlabs Cat# PM100D).

The multiphoton laser scanning microscope (FV-MPE-RS, Olympus Corporation), equipped with 25X water immersion objective (Olympus Corporation, Cat# XLPLN25XWMP2), was used for monitoring the fluorescence of GCaMP6f. The recordings began 10 s before stimulation and continued 20 s after stimulation for a total of 1 min. GCaMP6f proteins were excited with a 920 nm laser (Spectra-Physics Insight DL Dual-OL, Newport Corporation). Images were taken at 10 Hz with a $256 \times 256$ pixel resolution.

Acquired images were converted and analyzed in Fiji with the Olympus ImageJ plugin (http://imagej.net/OlympusImageJPlugin). Imaging windows were chosen that maximally captured the dense set of projections that emanate from Tk-GAL4$^{FruM}$ neurons before they branch out further (lateral junction described in *Yu et al., 2010*; *Figure 6B*). Rectangular regions of interest were selected and ΔF/F was calculated using a custom-written MATLAB code. First, the baseline fluorescence value ($F_{base}$) was calculated by averaging the fluorescence for 5 s preceding the stimulation. ΔF/F for each frame ($ΔF/F_{frame=N}$) was calculated as follows:

$$(\Delta F/F_{frame=N}) = [(F_{frame=N}) - -F_{base}]/F_{base}$$

Then, the $ΔF/F_{frame=N}$ for frames taken during the 30 s LED stimulation (excluding frames where the light was on) were averaged to calculate the ΔF/F of a given trial. Boxplots of average ΔF/F during stimulation at different LED powers and across flies were generated. Heterozygous controls and fruMB mutants' fluorescence data were compared across LED powers using the Mann-Whitney U-test ('ranksum').

## Acknowledgements

We thank Drs. David Anderson, Gerald Rubin, and Barret Pfeiffer for sharing unpublished transgenic *Drosophila* strains with us; Drs. Stephen Goodwin, Daisuke Yamamoto, and Anne von Philipsborn for other *Drosophila* strains; Eyrun Eyjolfsdottir and Dr. Pietro Perona for developing and improving the FlyTracker program; Dr. Michael Perry for sharing with us anti-DsxM and anti-FruM antibodies; Dr. Yong Wan for development and support for FluoRender; Dr. William Joiner for his help on circadian rhythm and sleep assays; Pavan Nayak and Vivian Shaw for their help on the development of behavior classifiers; Dr. Samuel Pfaff for sharing the Olympus FV-1000 confocal microscopy with us; Drs. Eiman Azim, Weizhe Hong, Samuel Pfaff, Carla Shatz, John Thomas, and members of the Asahina lab for critical comments on the manuscript, and David O'Keefe for scientific editing on the manuscript. The antisera nc82 (anti-BRP), developed by E Buchner, was obtained from the Developmental Studies Hybridoma Bank, created by the NICHD of the NIH and maintained at The University of Iowa, Department of Biology, Iowa City, IA 52242. Stocks obtained from the Bloomington *Drosophila* Stock Center (NIH P40OD018537) were used in this study. This work was also made possible in part by software funded by the NIH: FluoRender: Visualization-Based and Interactive Analysis for Multichannel Microscopy Data, 1R01EB023947-01 and the National Institute of General Medical Sciences of the National Institutes of Health under grant number P41GM103545-18.

## Additional information

### Funding

| Funder | Grant reference number | Author |
| --- | --- | --- |
| National Institute of General Medical Sciences | GM119844 | Kenichi Ishii Kenta Asahina |
| National Institute on Deafness and Other Communication Disorders | DC015577 | Margot Wohl Kenta Asahina |
| Naito Foundation | | Kenichi Ishii |

| Japan Society for the Promotion of Science | Kenichi Ishii |
| Mary K. Chapman Foundation | Margot Wohl |
| Rose Hills Foundation | Margot Wohl |

The funders had no role in study design, data collection and interpretation, or the decision to submit the work for publication.

### Author contributions
Margot Wohl, Software, Investigation; Kenichi Ishii, Investigation; Kenta Asahina, Conceptualization, Resources, Data curation, Formal analysis, Supervision, Funding acquisition, Validation, Investigation, Visualization, Methodology, Project administration

### Author ORCIDs
Kenta Asahina (iD) https://orcid.org/0000-0001-6359-4369

### Decision letter and Author response
Decision letter https://doi.org/10.7554/eLife.52702.sa1
Author response https://doi.org/10.7554/eLife.52702.sa2

## Additional files

### Supplementary files
• Source data 1. Data points represented in all figure panels, as well as p-values for all statistical tests shown in the figures.

• Supplementary file 1. Key resources table.

• Transparent reporting form

### Data availability
All data and statistical results are available in Source data 1.

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
