## [Decision Letter]

**Acceptance summary:**

The study analyses male-aggression, a sexually dimorphic behaviour. It shows that *Fruitless* function in a distinct subset of Tk-Gal4 positive neurons is required for male-male aggression independent of *Doublesex*, which acts in different subset of neuron to enable such aggression to be targeted towards male competitors and not females. The observation that the underlying circuits are different from those involved in courtship has interesting evolutionary implications.

**Decision letter after peer review:**

[Editors’ note: the authors submitted for reconsideration following the decision after peer review. What follows is the decision letter after the first round of review.]

Thank you for submitting your work entitled "A genetic code that specifies sexually dimorphic circuits and their control of social behaviors" for consideration by *eLife*. Your article has been reviewed by a Senior Editor, a Reviewing Editor, and three reviewers. The reviewers have opted to remain anonymous.

Our decision has been reached after consultation between the reviewers. In the discussions, all the referees expressed interest in the main message (dissociating *fru* and *dsx* functions and relating these to vertebrate mechanisms) as well as enthusiasm for several very interesting and clear findings from an impressively extensive set of experiments. However, there were also substantial concerns leading to the consensus opinion that additions and revisions required were too extensive to be accomplished in the two months required by *eLife*.

Based on these discussions and the individual reviews appended below, we regret to inform you that, in present form, your work will not be considered further for publication in *eLife*. However, if you choose to revise the paper on the lines suggested by the referees and resubmit, then *eLife* would be happy to consider a fresh submission (or two) that we would endeavour to have reviewed by the same referees.

In addition to the specific comments below, the referees offered some specific overall suggestions for revision that you may find useful. These include: (a) to refocus the paper on solely on the *fru*/*dsx* story; (b) to include new rigorous anatomical analysis of the "P1" cluster in P1^a^ and NP2631 flies as requested in the reviews; and (c) providing a detailed discussion of limitation of using systemic/constitutive alleles of *fru*/*dsx* in relation to behavior as well as anatomy. The referees also felt that the story be clarified and better communicated by splitting into two separate but simpler manuscripts, with a potentially a separate manuscript on Tk, but you should of course make your own decision on this rather too interventional opinion.

*Reviewer #1:*In both flies and mammals, multiple sexual differentiation mechanisms guide display of adult social behaviors. In flies it is *dsx* and *fru*, whereas in mammals it is largely testosterone and estrogen. Ishii et al. present a wide-ranging set of observations that they use to suggest that various aspects of sexually dimorphic courtship and aggression displays in flies are controlled by either *dsx* or *fru*. This seems an over-interpretation of the findings. I am supportive of the study but have a few suggestions.

1) Data from Figure 3 is used to conclude that *dsx* specifies both P1 and NP2631 neurons. This is inconsistent with the anatomy shown in figure panels, which shows that both *dsx* and *fru* contribute. It is true that cell number is dependent on *dsx*, but the morphology of the arbors indicates that *fru* is essential as well: E3/J3/K3 are different than C3/H3/I3. Similarly, data from Figure 6 is used to conclude that fruM but not *dsx* specifies Tk neurons; comparison of E3 and F3 indicates that dsxM is also important. The authors should quantify arborization rather than rely on gross examination for Figure 3 and Figure 6.

Presentation of the histology could be better elsewhere too. For example, from panels Figure 2A,C, I conclude that there is no overlap of *R15A01* and *R71G01*, raising the question of the intersectional identification of P1^a^ neurons.

2) It is clear that FruM is essential for male type lunges toward other males and enhancing wing extension to females. However, FruM is also essential for the pattern of wing extension upon activation of P1^a^ neurons; in controls (Figure 1), wing extensions in phase 2>3 and 2<3 toward males and females, whereas in FruF males (figure 4A-D), wing extensions in phase 2<3 toward males and females. In addition, it is clear that FruM is essential for regulating intensity of wing extensions. It is a gross simplification to conclude that dsxM enables execution of male courtship whereas FruM enforces choice of target sex. Clearly, FruM is also critical for execution of this behavior.

3) Why do males not lunge toward other males in the absence of activation of P1^a^ or NP2631 or Tk neurons (Figure 1, Figure 6)?

4) There is some over-simplification of the literature. It might be true in flies that most studies have only used one sex as a target individual to analyze function of particular neurons (subsection “Effect of a target fly’s sex on optogenetically induced social interactions”), but this is certainly not true in mice (studies from Anderson, Dulac, Lin, Shah).

Rather than draw parallels to the four core genotype in the mouse (using which not a single gene has been identified that regulates social behavior independent of sex hormones, subsection “*dsx*, and not *fru*, specifies both P1^a^ and NP2631 ∩ dsx^FLP^ neurons”), it would be more appropriate to discuss the role of estrogen vs. testosterone in regulating male behaviors. It is clear that estrogen masculinizes the brain during a critical window whereas testosterone signaling amplifies male behaviors in adult life. This would be a more instructive analogy when trying to dissect the specific functions of *dsx* and *fru*.*Reviewer #2:*Unfortunately, I have some major technical concerns with the manuscript, and the lack of quantitative anatomical analyses throughout. The rather cursory analysis offered in this manuscript has several critical caveats that are not discussed and undermine the conclusions that can be drawn from the experimental data. Therefore, this manuscript is not suitable for publication without significant experimental clarification of these points.

Subsection “Effect of a target fly’s sex on optogenetically induced social interactions” and throughout the manuscript:

Neither the P1^a^ split-Gal4 nor NP2631 ∩ dsx^FLP^ driver are 'P1/pC1'-specific, some additional neurons in the brain express Gal4 in both cases (see both Hoopfer et al., 2015 and Koganezawa et al., 2016). Therefore, by using these drivers, the authors cannot conclude that the observed behavioral phenotypes were caused purely by the purported 'P1/pC1' neurons. Further intersectional approaches to rigorously target these neurons are essential.

Subsection “*dsx*, and not *fru*, specifies both P1^a^ and NP2631 ∩ dsx^FLP^ neurons”:

The expression 'P1^a^ neurons were specified only in the flies that had dsxM' is not accurate and is likely even wrong. Whether the presence of the P1 neurons depends on *fru* or *dsx* was originally examined in the very first P1 paper, Kimura et al., 2008. Kimura et al. previously demonstrated by using *dsx* mutations that the absence of dsxF, but not the presence of dsxM, is the most striking factor for determining the presence of the P1 neurons.

In Figure 3, the *dsx* locus was always wild type and therefore the authors could not dissociate the presence of dsxM and the absence of dsxF. It is essential for the authors to do the similar experiments in combination with the appropriate *dsx* mutant alleles.

The authors mentioned that 'processes of P1^a^ neurons in fruF males appeared thinner than those in males'. What does thinner mean? What form of scientific measurement is "thinner"? This needs to be quantified properly. Image co-registration or any other quantitative analyses for the size differences of the neuronal processes between these genotypes must be carried out.

The authors also mention ‘sex differences both in terms of cell body numbers and branching patterns’. If the authors declare the branching patterns were different between genotypes, they must carry out quantitative analyses.

They also declare the neuronal morphology was 'similar' between particular genotypes, but what does 'similar' mean? The authors seem to conclude that the neuronal morphology is different when they expect it is different and that it is similar when they expect it is similar, without any quantitative criteria for measurement.

Subsection “P1^a^ and NP2631 ∩ dsx^FLP^ neurons in fruF males can induce courtship, but not male-type aggressive behaviour”:

The logical link between the anatomy in Figure 3 and behavior in Figure 4 is unclear. In Figure 3, the authors looked at the contributions of the *fru* and *dsx* genes in the purported 'P1/pC1'-specific neurons, but in Figure 4, they carried out behavioral assays with the *fru* mutants, in which *fru* gene function is disrupted not only in P1/pC1 neurons but also all other FruM-expressing neurons throughout the PNS and CNS, they then base all their conclusions on only the behavioral function of P1/pC1 neurons. These animals will have a plethora of *fru* mutant phenotypes that are associated with lack of FruM in other neurons in the CNS and PNS.

To make any relevant conclusions the authors need to re-do the behavioral assays in Figure 4 using the flies in which only the P1/pC1 neurons are mutant for *fru* (e.g., mutant MARCM clones, UAS-Cas9 system, or fru-RNAi).

Subsection “fruM specifies aggression-promoting Tk-GAL4^FruM^ neurons”:

The logic of transition from P1/pC1 to Tk-Gal4 neurons is very difficult to follow. The relationship between the former and the latter need to be explained. Are they completely different stories?

Figure 6:

As a control, the authors should demonstrate that Tk-GAL4 neurons do not express *dsx*.

Subsection “Differential roles of *fru* isoforms on male-male interactions and specification of Tk-GAL4^FruM^ neurons”:

The authors showed that the Tk-Gal4 neurons exist in *fru*^∆B^ mutant males but behavior is affected in these flies. The simplest explanation would be that the neurites are malformed in these flies.

Although the authors mentioned that 'fruMA and fruM mutants did not affect […] their arborization pattern', they didn't seem to carry out any quantitative morphological analyses. Such analyses should be included.

Subsection “The Tk-GAL4^FruM^ neurons in fruMB mutants can induce male aggression”:

The authors mention that 'these mutants have Tk-GAL4^FruM^ neurons that appear to retain their morphology'. But how did they examine this?

*Reviewer #3:*This paper is a paradox. As far as I can tell it is rigorous on all levels. The scholarship is excellent, deep and well rounded. In addition, the issues are explored thoroughly and there is an enormous amount of data. In addition, an enormous number of fly lines (the table that describes the lines is phenomenal). But on the other hand, it is too much. I would like to discuss the argument and conclusions that surface relating *dsx* and *fru* expressing neurons with sexually dimorphic choice vs. sexually dimorphic motor patterns and the interesting differences between reproductive behavior and aggression, *but* the paper is simply too dense at almost every point. Reading it is a painful experience and this does the work a tremendous disservice. I say this recognizing and admiring the care that has gone into preparing this text. It seems flawless, just impossibly dense.

What to do? Here is a list of suggestions.

1) The text and figures are impeccably constructed but for the average reader they are overly complicated and far too dense. The flow of the manuscript would benefit greatly from severe editing of the text, figures and figure legends. It is with a heavy heart that I ask the authors to do this knowing the amount of effort it took to construct the manuscript.

2) Remove raster plots from figures throughout and move to supplemental material if needed. See Figure 1, Figure 4, Figure 5, Figure 6, Figure 7, Figure 9.

3) Figure 2 is not required. Move to supplemental material and concisely state result in text.

4) Much of the immunohistochemistry could be removed from figures and moved to supplemental material. See Figure 3, Figure 6, Figure 8.

5) The summary panels at the end of some of the figures are helpful. But a single unifying summary/model would be best. Though organized in a systematic and rational way the results still feel piecemeal and it is tiresome to continually have refer back to previous figures or supplemental material to regain your bearing. A single model would help immensely.

Alternatively, it might make sense to turn this into two papers, one about sex, the other about aggression?

[Editors’ note: further revisions were suggested prior to acceptance, as described below.]

Thank you for submitting your article "Layered roles of *fruitles*s isoforms in specification and function of male aggression-promoting neurons in *Drosophila*" for consideration by *eLife*. Your article has been reviewed by K VijayRaghavan as the Senior Editor, a Reviewing Editor, and two reviewers. The reviewers have opted to remain anonymous.

The reviewers have discussed the reviews with one another and the Reviewing Editor has drafted this decision to help you prepare a revised submission.

Summary:

In this follow up paper, the authors analyse male-aggression, a different sexually dimorphic behaviour and that *fru* function in a distinct subset of Tk-Gal4 positive neurons is required for male-male aggression independent of *Doublesex*, which acts in different subset of neuron to enable such aggression to be targeted towards male competitors and not females. The observation that the underlying circuits are different from those involved in courtship has interesting evolutionary implications.

Essential revisions:

The issues enumerated below need to be clarified before the paper can be accepted.

1) That activation of Tk-Gal4 neurons in single-housed FruM females does not elicit aggression is at odds with the conclusion that this circuit is sufficient for executing male type aggression. If anything, single housing should have increased aggression. The authors' suggest that the lack of aggression is because of increase in male-type courtship, but this still tacitly indicates that activating the Tk/FruM circuit is insufficient to induce aggression. The authors' conclusions should be tempered accordingly.

2) As with the companion paper (“Sex-determining genes distinctly regulate courtship capability and preference toward females through sexually dimorphic neurons”), it would appear that the effects of Fru isoform on aggression in Tk-GAL4^FruM^ neurons also can only really be assessed by looking at animals were the Tk-GAL4^FruM^ neurons are mutant/ perturbed for *fru* function (e.g. *fru* isoform-specific mutant MARCM clones, UAS-Cas9 system, or fru-RNAi). Otherwise, the data would need to be interpreted with the caveat that *fru* mutant phenotypes could be caused by lack of Fru isoform in other neurons in the CNS and/or the PNS. This issue needs to addressed directly – ideally experimentally given that a lineTK-GAL4/fruP1LexA/LexAop2-FLPL that may allow these neurons to be targeted has been described in Asahina et al., 2014.

3) A key finding arising from optogenetic experiments, artificially activating Tk-GAL4^FruM^ neurons are a refinement of the authors previously published thermogenetic work (Asahina et al., 2014), and the isoform-specific roles of Fru on male behavior and Fru-specified neural circuits, are implied by work from other groups (Neville et al., 2014; von Philipsborn et al., 2014). The discussion must clearly articulate what new information this manuscript provides on Fru isoform function and aggressive behavior.

4) Does Figure 2—figure supplement 2E, F_1_ (subsection “Tk-GAL4^FruM^ neurons are a part of a fruM-dependent circuit for the execution of male-type aggressive behaviour”) refer to a different figure panel. Please correct.

5) *Dsx* immunolabeling in Figure 1—figure supplement 2C is poor. Please provide better images.

---

## [Author Response]

[Editors’ note: the authors resubmitted a revised version of the paper for consideration. What follows is the authors’ response to the first round of review.]

Our decision has been reached after consultation between the reviewers. In the discussions, all the referees expressed interest in the main message (dissociating fru and dsx functions and relating these to vertebrate mechanisms) as well as enthusiasm for several very interesting and clear findings from a impressively extensive set of experiments. However, there were also substantial concerns leading to the consensus opinion that additions and revisions required were too extensive to be accomplished in the two months required by eLife.Based on these discussions and the individual reviews appended below, we regret to inform you that, in present form, your work will not be considered further for publication in eLife. However, if you choose to revise the paper on the lines suggested by the referees and resubmit, then eLife would be happy to consider a fresh submission (or two) that we would endeavour to have reviewed by the same referees.In addition to the specific comments below, the referees offered some specific overall suggestions for revision that you may find useful. These include: (a) 1 to refocus the paper on solely on the fru/dsx story; (b) to include new rigorous anatomical analysis of the "P1" cluster in P1a and NP2631 flies as requested in the reviews; and (c) providing a detailed discussion of limitation of using systemic/constitutive alleles of fru/dsx in relation to behavior as well as anatomy. The referees also felt that the story be clarified and better communicated by splitting into two separate but simpler manuscripts, with a potentially a separate manuscript on Tk, but you should of course make your own decision on this rather too interventional opinion.

We decided to split our original manuscript into two, following two reviewers’ suggestions that this would increase clarity. We agree that our attempts to integrate our findings on 2 genes (*dsx* and *fru*), 2 behaviors (courtship and aggression), and 3 neuronal populations (NP2631 ∩ dsx^FLP^, P1^a^ , and Tk-GAL4^FruM^ neurons) caused difficulties with the flow of logic. We believe that two separate manuscripts allow us to focus on two relatively simpler messages individually, and help us present relevant data without causing confusion. “Sex-determining genes distinctly regulate courtship capability and target preference via sexually dimorphic neurons” also serves to incorporate your first major recommendation: “to refocus the paper on solely on the *fru*/*dsx* story”. In line with this recommendation, we now include a new set of data showing that *dsx* and *fru*-co-expressing neurons in males can promote both courtship and aggressive behaviors in a target sex-dependent manner (see Figure 1 of that manuscript).

Regarding your second recommendation to provide rigorous neuroanatomical analysis, we thoroughly re-evaluated our anatomical data by registering brains to a common template and analyzing specific volumes with 3-dimentional segmentation. Please see Figure 3L-M, Figure 6H, I, in “Sex-determining genes distinctly regulate courtship capability and target preference via sexually dimorphic neurons”, and Figure 1N-P, and Figure 5I-K in “Layered roles of*fruitless*isoforms in specification and function of male aggression-promoting neurons in *Drosophila*” for new quantification of neuronal morphologies. Standardized immunohistochemical data enabled us to directly compare the morphology of specific neurons across multiple genotypes. Moreover, we could detect statistical differences in neuroanatomy by quantitatively comparing segmented volumes of specific neuronal structures. This rigorous quantification revealed differences in P1^a^ neurons between males and fruF males, which compelled us to revise our original conclusions for this class of neurons.

In response to your third recommendation, we added a discussion of the limitations of studying sex-determining genes through constitutive mutants and through cell-specific genetic manipulation (see “An organismal sex and a cellular sex” section in Discussion of “Sex-determining genes distinctly regulate courtship capability and target preference via sexually dimorphic neurons”). We believe that cell-specific manipulation of sex-determining genes is a powerful method that is not without weaknesses, and will serve complimentary roles to constitutive mutants when studying their effects on neuroanatomy and behaviors.

Reviewer #1:In both flies and mammals, multiple sexual differentiation mechanisms guide display of adult social behaviors. In flies it is dsx and fru, whereas in mammals it is largely testosterone and estrogen. Ishii et al. present a wide-ranging set of observations that they use to suggest that various aspects of sexually dimorphic courtship and aggression displays in flies are controlled by either dsx or fru. This seems an over-interpretation of the findings. I am supportive of the study but have a few suggestions.1) Data from Figure 3 is used to conclude that dsx specifies both P1 and NP2631 neurons. This is inconsistent with the anatomy shown in figure panels, which shows that both dsx and fru contribute. It is true that cell number is dependent on dsx, but the morphology of the arbors indicates that fru is essential as well: E3/J3/K3 are different than C3/H3/I3. Similarly, data from Figure 6 is used to conclude that fruM but not dsx specifies Tk neurons; comparison of E3 and F3 indicates that dsxM is also important. The authors should quantify arborization rather than rely on gross examination for Figure 3 and Figure 6.

We agree that we should have quantified our findings to draw conclusions about the neuroanatomy of these neurons. Reviewer #2 also raised concerns about our anecdotal statements regarding neuroanatomy. We therefore thoroughly re-analyzed our anatomical data by registering brains to a “template” *Drosophila* brain, and by using 3-dimentional segmentation and volume measurements. Please see Materials and methods section for technical details. In essence, we used a non-rigid transformation technique described by Jefferis et al., 2007, to register individual sample brains to the standard unisex *Drosophila* brain described in Bogovic et al., 2018. This allowed us to compare labeling patterns across multiple brains and sexes in the single reference space.

Immunohistochemical labeling can have technical as well as biological variability. Such inter-sample variability might have been perceived as inter-genotype differences. To better represent labeling patterns that are consistent across samples, we averaged the signal intensity of imaged neurons after registration in the standard brain. The resulting images are shown in 2-dimensional projections as well as in 3-dimensional movies. A similar approach was taken to show the neuroanatomy of various *fru*-expressing neurons (Yu et al., 2010, von Philipsborn et al., 2014), and we feel this is the best approach to concisely summarize our observations from multiple samples.

As a quantification, we compared the volumes of specific neuronal innervations across different genotypes. We used the 3-dimensional rendering program FluoRender to segment structures of interest, and calculated the volumes of each structure.

With these new and rigorous approaches, we are now able to quantitatively support our initial conclusions that (1) NP2631 ∩ dsx^FLP^ neurons are largely specified by *dsx*, and (2) Tk-GAL4^FruM^ neurons are specified by *fru*.

In “Sex-determining genes distinctly regulate courtship capability and preference toward females through sexually dimorphic neurons” Figure 3H-K and Video 3, Video 4 and Video 5, we demonstrate that the overall morphology of NP2631 ∩ dsx^FLP^ neurons in males and fruF males overlap well, and that these neurons in females and fruM females are also similar. We quantified three neuropils that show noticeable sexual dimorphism, and in all cases, volumes of these neuropils in fruF males are statistically indistinguishable from those in males. Volumes in fruM females are comparable to values in females except in one neuropil. We acknowledge that our quantification is based on the entire neural population, and that we may not have sufficient sensitivity to detect either subtype-specific contributions of *fru*, or differences at finer scales such as number or length of branches. We mention this possibility in the main text. Overall, however, our analysis provided us with evidence arguing that sexual dimorphism of NP2631 ∩ dsx^FLP^ neurons is mainly, if not entirely, specified by *dsx*.

Likewise, data in “Layered roles of *fruitless* isoforms in specification and function of male-type aggression-promoting neurons in *Drosophila*” Figure 1L, M and Video 3 clearly show that average Tk-GAL4^FruM^ neurons from males and from fruM females almost perfectly overlap (light cyan in Video 3 indicates the overlap between male (green) and fruM female (dark blue) Tk-GAL4^FruM^ neurons). As is shown in “Layered roles of *fruitless* isoforms in specification and function of male-type aggression-promoting neurons in *Drosophila*” Figure 1N-P, three prominent innervations of these neurons show similar volumes across males (both *fru* +/+ and *fru^M^/fru^4-40^*) and fruM females. We therefore retain our original conclusion that Tk-GAL4^FruM^ neurons are predominantly specified by *fru*.

In contrast, we found that both *dsx* and *fru* are important for specifying P1^a^ neurons. In the previous version, we hastily concluded that *dsx* specified P1^a^ neurons mainly because P1^a^ neurons appear in fruF male brains, but not in fruM female brains. However, quantification of volumes of two neuropils clearly indicates that P1^a^ neurons in fruF males have distinct morphology from those in males (see “Sex-determining genes distinctly regulate courtship capability and preference toward females through sexually dimorphic neurons”, Figure 6F-I). Results from this and re-analysis of behavioral data (discussed below) compelled us to revise our conclusion about P1^a^ neurons. We now fully acknowledge the role of *fru* in both specification and function of P1^a^ neurons. We feel this is a striking contrast to NP2631 ∩ dsx^FLP^ neurons, and further supports the idea that these two neurons are distinct subpopulations within what has been collectively described as P1 or pC1 neurons.

We thank the reviewer for encouraging us to take an extra step for verification (which turned out to require substantial manual and computational works), as we feel our revised conclusions are now better supported by the data.

Presentation of the histology could be better elsewhere too. For example, from panels Figure 2A,C, I conclude that there is no overlap of R15A01 and R71G01, raising the question of the intersectional identification of P1a neurons.

We admit that images in our original version were not sufficiently clear, especially on printed media. It is a challenge to show all visible neurons clearly without over-saturating strongly labeled cells, or without dominating the figure panel. We enlarged the images by 20%, and would like to draw attention to original images provided as source data.

Regarding the overlap of R15A01 and R71G01 promoters, our initial submission used the *R71G01-LexA* transgene in attP40. It is known in the community that transgene expression at the attP40 landing site is not as robust as at the attP2 landing site, which is used for creating the Rubin GAL4 collection. We had created a transgenic animal with *R71G01-LexA* in attP2, and tested its overlap with NP2631 ∩ dsx^FLP^ and with NP2631 ∩ fru^FLP^ (“Sex-determining genes distinctly regulate courtship capability and preference toward females through sexually dimorphic neurons”, Figure 5C, D and Figure 5—figure supplement 3C, D, respectively). As anticipated, we observed more *R71G01-LexA*-labeled neurons from the attP2 transgene than from the attP40 transgene. The overlap with NP2631 ∩ dsx^FLP^ or NP2631 ∩ fru^FLP^ neurons is still very small, confirming our initial conclusion that P1^a^ neurons have minimal overlap with these neurons. We strengthen this conclusion by comparing registered and averaged 3D images of NP2631 ∩ dsx^FLP^ and P1^a^ neurons (“Sex-determining genes distinctly regulate courtship capability and preference toward females through sexually dimorphic neurons”, Figure 5E-G). Neuroanatomy of these two populations shows significant differences.

In addition, Hoopfer et al., 2015 showed that optogenetic stimulation of *R15A01-LexA* neurons recapitulates the behavioural effects of the P1^a^ neuronal activation (see Figure 3H in the citation). This data suggest that *R1501-LexA* indeed includes P1^a^ neurons. We therefore remain confident with our conclusion which this piece of data is most relevant: NP2631 ∩ dsx^FLP^ and P1^a^ neurons are largely separate subpopulations.

We would like to add that split GAL4 lines are often created from lines labeling seemingly different neurons at a population level, especially in z-projection images. In fact, it may be difficult to discern overlapping neurons from the images of R15A01-GAL4 and R71G01-GAL4 in Figure 1C of Hoopfer et al., 2015 (which described

P1^a^ neurons).

2) It is clear that FruM is essential for male type lunges toward other males and enhancing wing extension to females. However, FruM is also essential for the pattern of wing extension upon activation of P1a neurons; in controls (Figure 1), wing extensions in phase 2>3 and 2<3 toward males and females, whereas in FruF males (Figure 4A-D), wing extensions in phase 2<3 toward males and females. In addition, it is clear that FruM is essential for regulating intensity of wing extensions. It is a gross simplification to conclude that dsxM enables execution of male courtship whereas FruM enforces choice of target sex. Clearly, FruM is also critical for execution of this behavior.

The reviewer was right in pointing out the importance of *fru* on the function, as well as on neuroanatomy (discussed above), of P1^a^ neurons. We confirmed that optogenetic stimulation of P1^a^ neurons in fruF males induced significantly less wing extensions than in regular males regardless of target sex (“Sex-determining genes distinctly regulate courtship capability and preference toward females through sexually dimorphic neurons”, Figure 6—figure supplement 2). This means that the courtship-executing function, as well as specification, of P1^a^ neurons require the activity of both *dsx* and *fru*. Currently, we do not know whether *dsx* and/or *fru* is necessary within P1^a^ neurons, or whether the proper function of P1^a^ neurons requires another population of *dsx*- or *fru*-dependent neurons (see also discussions regarding cell type-specific manipulation of sex-determining genes). We therefore reasoned that it would be prudent to avoid speculations about the roles of *dsx* and *fru* on P1^a^ neurons, and decided to re-frame P1^a^-related data in the context of discussing the potential diversity of the P1/pC1 neuronal cluster.

On the other hand, optogenetic stimulation of NP2631 ∩ dsx^FLP^ neurons in fruF males induced a similar amount of wing extensions as the same manipulation in regular males did toward male target flies (“Sex-determining genes distinctly regulate courtship capability and preference toward females through sexually dimorphic neurons”, Figure 4C). Together with our observation that the neuroanatomy of NP2631 ∩ dsx^FLP^ neurons in regular males and in fruF males showed little difference, we conclude that (1) fruF males are capable of executing wing extensions, (2) NP2631 ∩ dsx^FLP^ neurons are one neuronal component that belongs to a *dsx*-dependent wing extension execution circuit, and (3) fruM is important for enhancing courtship behavior specifically toward females.

By referring to the “execution” of courtship behavior, we are specifically focusing on the animal’s capability to express courtship behavior. We aim to uncover a neurogenetic process that enables animals to perform courtship behavior at all, in contrast to the ability to adjust the intensity of courtship behavior toward females. Please note that we *do not* argue that *fru* is not required for proper courtship behavior. What we argue is that *fru* is not necessary for *all* aspects of male courtship behavior. The importance of *fru* on male sexual behavior has been extensively demonstrated, leading some to state that fruM specifies all aspects of male sexual behaviors. Here, we wish to shed light on the fundamental characteristics of *fru* mutants, which is the unusually high level of courtship behavior toward males, not the absence of courtship behavior altogether. This phenotype has been consistently and repeatedly reported (literature include Gailey and Hall (1989), Villella et al. (1997), Ito et al. (1996), Demir and Dickson (2005), Shirangi et al. (2006), Pan and Baker (2014), among others), even though these observations may not have received sufficient attention. Moreover, many of previously characterized *fru*-expressing neurons are sensory neurons or early-stage interneurons that transmit sensory information. We feel that our conclusions are entirely consistent with past literature. Our novel functional characterization of NP2631 ∩ dsx^FLP^ neurons provides a neuronal substrate that accounts for the courtship behavior-execution capability of fruF males.

3) Why do males not lunge toward other males in the absence of activation of P1a or NP2631 or Tk neurons (Figure 1, Figure 6)?

The tester flies were all housed in groups for 6 days (P1^a^ neurons) or 14 days (for NP2631 ∩ dsx^FLP^ neurons), which potently reduces spontaneous aggressive behavior. Socially isolated animals show high levels of spontaneous aggression, which can cause a ceiling effect when trying to observe further increases of aggression. Both in flies and mice, group-housing is a common treatment to characterize behavioral changes in response to neuronal manipulations.

4) There is some over-simplification of the literature. It might be true in flies that most studies have only used one sex as a target individual to analyze function of particular neurons (subsection “Effect of a target fly’s sex on optogenetically induced social interactions”), but this is certainly not true in mice (studies from Anderson, Dulac, Lin, Shah).Rather than draw parallels to the four core genotype in the mouse (using which not a single gene has been identified that regulates social behavior independent of sex hormones, subsection “dsx, and not fru, specifies both P1a and NP2631 ∩ dsxFLP neurons”), it would be more appropriate to discuss the role of estrogen vs. testosterone in regulating male behaviors. It is clear that estrogen masculinizes the brain during a critical window whereas testosterone signaling amplifies male behaviors in adult life. This would be a more instructive analogy when trying to dissect the specific functions of dsx and fru.

We appreciate the reviewer’s constructive suggestion to re-frame the roles of *dsx* and *fru*. Indeed, studies in mice often use both males (often castrated) and females as intruders. While that these neuronal manipulations may appear to trigger social behaviors largely independent of target sex, we found that some target sex-dependence has been previously reported. For instance, Lin et al. (2010) reported that optogenetic stimulation of the mediolateral part of the ventromedial hypothalamus (VMHml) triggered aggression more consistently toward males than toward females (Figure 4L,M in the citation). Likewise, close examination of data from Yang et al. (2017), which investigated the behavioral changes caused by chemogenetic activation of progesterone receptor expressing neurons in VMHml, suggests that aggression toward males was more consistent that toward females, although it was not explicitly tested (Figure 1F,G in citation). These past observations reinforce our idea that target sex in neuronal manipulation experiments can be an important variable in other experimental models.

We also agree that the “four core genotypes” model has not been particularly successful in providing mechanical insight into the origin of sexual dimorphism. Although the temporal dynamics between mammalian sex hormones and *Drosophila* sex-determining genes can be quite different, the reviewer’s opinion that the roles of *dsx* and *fru* can be analogous to those of estrogen and testosterone is indeed a quite useful framework for our study. We incorporated this suggestion into our Discussion.

Reviewer #2:Unfortunately, I have some major technical concerns with the manuscript, and the lack of quantitative anatomical analyses throughout. The rather cursory analysis offered in this manuscript has several critical caveats that are not discussed and undermine the conclusions that can be drawn from the experimental data. Therefore, this manuscript is not suitable for publication without significant experimental clarification of these points.

As discussed in response to reviewer #1, we incorporated quantification of our neuroanatomical data. Additional details are provided below, and we hope these new analyses address the reviewer’s concerns.

Subsection “Effect of a target fly’s sex on optogenetically induced social interactions” and throughout the manuscript:Neither the P1a split-Gal4 nor NP2631 ∩ dsxFLP driver are 'P1/pC1'-specific, some additional neurons in the brain express Gal4 in both cases (see both Hoopfer et al., 2015 and Koganezawa et al., 2016). Therefore, by using these drivers, the authors cannot conclude that the observed behavioral phenotypes were caused purely by the purported 'P1/pC1' neurons. Further intersectional approaches to rigorously target these neurons are essential.

We are a little confused by the reviewer’s assertion that “Neither P1^a^-GAL4 or NP2631 ∩ dsx^FLP^ are ‘P1/pC1’-specific”, after citing the previous publications. In both cases, we see in citation figures that these drivers in fact label a single cluster of neurons cleanly, both in original publications as well as in our own studies.

For P1^a^-GAL4 neurons, Figure 1E of Hoopfer et al., 2015 shows clearly that only a single cluster of neurons at the posterior end of a male brain were labeled. We confirmed this specificity repeatedly, as shown in “Sex-determining genes distinctly regulate courtship capability and preference toward females through sexually dimorphic neurons”, Figure 6A, Figure 6—figure supplement 1A, C. NP2631 ∩ dsx^FLP^ neurons described in Figure 4D of Koganezawa et al., 2016 also seems to label a well-defined single cluster of neurons per hemisphere. Again, we observed a bilateral single cluster of neurons at the posterior edge of male brains (“Sex-determining genes distinctly regulate courtship capability and preference toward females through sexually dimorphic neurons”, Figure 3C, Figure 3—figure supplement 1A, C). Please also see the movies showing the 3-dimentional structure of P1^a^ and NP2631 ∩ dsx^FLP^ neurons.

We wondered whether the reviewer was raising the possibility that this seemingly single cluster of neurons may indeed consist of heterogeneous populations. Although the number of cells that are labeled by both drivers is rather small (“Sex-determining genes distinctly regulate courtship capability and preference toward females through sexually dimorphic neurons”, Figure 3G and Figure 6E), further intersection could reveal a specific subset that is responsible for the observed behaviors. We currently do not have a means to reliably separate such subpopulations, and we also feel that such experiments would better belong to a future study. Here, we focus on the existence of *dsx*-dependent neurons that can execute wing extensions, and a separate, *fru*-dependent mechanism that enhances wing extensions toward females.

Subsection “dsx, and not fru, specifies both P1a and NP2631 ∩ dsxFLP neurons”:The expression 'P1a neurons were specified only in the flies that had dsxM' is not accurate and is likely even wrong. Whether the presence of the P1 neurons depends on fru or dsx was originally examined in the very first P1 paper, Kimura et al., 2008. Kimura et al. previously demonstrated by using dsx mutations that the absence of dsxF, but not the presence of dsxM, is the most striking factor for determining the presence of the P1 neurons.In Figure 3, the dsx locus was always wild type and therefore the authors could not dissociate the presence of dsxM and the absence of dsxF. It is essential for the authors to do the similar experiments in combination with the appropriate dsx mutant alleles.

We admit that we were careless in stating that either dsxM or dsxF is responsible for neuroanatomical or behavioral differences from regular males. With the reagents we used, we could only state whether sexually dimorphic splicing of *dsx* has an impact on NP2631 ∩ dsx^FLP^ or P1^a^ neurons. We do not know whether altered neuroanatomy of P1^a^ neurons in fruF males is due to the lack of dsxM, or due to the presence of dsxF. As the reviewer points out, we must use *dsx* mutants (especially dsx^Dom^, which forces male-type splicing of *dsx* regardless of sex chromosome composition) to gain insight into which of the two isoforms is responsible.

However, we would like to emphasize that it is not the specific role of dsxM or dsxF, but rather the genetic origin of sexual dimorphism, that we wished to investigate in this study. That is, we are interested in addressing which sexually dimorphic characteristics (neuroanatomical and behavioral) are controlled by the dimorphism (e.g., sex-specific splicing) of *dsx* and *fru*. As we state in the Discussion, one sex is not a mutant of the other sex, and sex-determining genes often have different functions in each of the two sexes – although *fru* was often considered silent in females, we presented circumstantial evidence that the fruF allele may not be truly null (“Layered roles of *fruitless* isoforms in specification and function of male-type aggression-promoting neurons in *Drosophila*”, Figure 1—figure supplement 3A-C). In this context, splicing mutants such as fru^M^ and fru^F^ provide a useful platform to understand the genetic nature of sex-specific transformation. These two alleles have the added benefit that experiments could be conducted in a symmetrical manner – we can exhaust all four possible combinations of *dsx* and *fru* sex-specific isoforms (“Sex-determining genes distinctly regulate courtship capability and preference toward females through sexually dimorphic neurons”, Figure 3B). Currently, no mutation that forces female-type splicing of *dsx* is known.

We believe our data present strong evidence that *dsx* and *fru* have distinct impacts on the specification and function of sexually dimorphic neuronal circuits. We revised our statements and referred to the importance of *dsx* as a gene, not of sex-specific splicing isoforms.

The authors mentioned that 'processes of P1a neurons in fruF males appeared thinner than those in males'. What does thinner mean? What form of scientific measurement is "thinner"? This needs to be quantified properly. Image co-registration or any other quantitative analyses for the size differences of the neuronal processes between these genotypes must be carried out.The authors also mention ‘sex differences both in terms of cell body numbers and branching patterns’. If the authors declare the branching patterns were different between genotypes, they must carry out quantitative analyses.They also declare the neuronal morphology was 'similar' between particular genotypes, but what does 'similar' mean? The authors seem to conclude that the neuronal morphology is different when they expect it is different and that it is similar when they expect it is similar, without any quantitative criteria for measurement.

As stated above in reply to a similar request from reviewer #1, we did an extensive re-analysis of our neuroanatomical data. We followed reviewer #2’s request, and hope that our new set of data will directly address the reviewers’ concerns.

Subsection “P1a and NP2631 ∩ dsxFLP neurons in fruF males can induce courtship, but not male-type aggressive behaviour”:The logical link between the anatomy in Figure 3 and behavior in Figure 4 is unclear. In Figure 3, the authors looked at the contributions of the fru and dsx genes in the purported 'P1/pC1'-specific neurons, but in Figure 4, they carried out behavioral assays with the fru mutants, in which fru gene function is disrupted not only in P1/pC1 neurons but also all other FruM-expressing neurons throughout the PNS and CNS, they then base all their conclusions on only the behavioral function of P1/pC1 neurons. These animals will have a plethora of fru mutant phenotypes that are associated with lack of fruM in other neurons in the CNS and PNS.To make any relevant conclusions the authors need to re-do the behavioral assays in Figure 4 using the flies in which only the P1/pC1 neurons are mutant for fru (e.g., mutant MARCM clones, UAS-Cas9 system, or fru-RNAi).

Our experiments address what the NP2631 ∩ dsx^FLP^ or P1^a^ neurons can do in fruF males. We would like to clarify that our original submission *did not* state that *fru* is required cell autonomously in either NP2631 ∩ dsx^FLP^ or P1^a^ neurons, precisely because we were aware that we could not attribute the behavioral differences from regular males to the deficit specifically within either neuronal population. We apologize if our wording in the previous submission caused any confusion.

We agree with the reviewer’s critique that the use of constitutive mutants has a limitation. We cannot answer whether the phenotype (anatomical or behavioral) is due to the lack of *dsx* or *fru* functions in the neurons of interest, or due to their cell non-autonomous functions (likely in other neurons). However, we would like to reiterate that the cell-autonomous function of *dsx* or *fru* is not the focus of this study. We wished to address how sexual dimorphism of *dsx* and *fru* (in a form of sex-specific splicing) may contribute to the specification of sexually dimorphic circuits and sexually dimorphic social behaviors. As stated above, simple loss-of-function of *dsx* or *fru* does not necessarily distinguish which male-like or female-like characteristics are specified by a given gene, because the loss of a gene is not synonymous to the transformation of sex from one to the other. We respectfully disagree with the reviewer’s assertion that we cannot make “any relevant conclusions” with these mutants. Rather, we believe that systematic investigation into what changes in fruF males and fruM females at circuit and behavioral levels (e.g., comparison of all 4 genotypes shown in “Sex-determining genes distinctly regulate courtship capability and preference toward females through sexually dimorphic neurons”, Figure 3B) is crucial for understanding the transformative nature of sex determination.

We would also like to be mindful of the limitations of cell-type specific manipulations of *dsx* and *fru*. For instance, successful gene knockdown using ‘dead Cas9’ (CRISPRi) crucially depends on the sequence of 20-base guide RNA. RNA interference (RNAi) relies on the specificity of ~21-base double-strand RNA sequence. In both cases, off-target effects as well as incomplete (or even ineffective) knockdown of the target gene are prevalent and well documented. Incomplete knockdown can be also caused by varied strengths of GAL4 drivers. To make matters more complicated, *dsx* and *fru* are likely crucial during development. This means that a GAL4 driver needs to turn on very early and remain active consistently during development to provide interpretable results. In previous studies, characterization of both the efficacy of *fru* RNAi knockdown or the temporal activity or GAL4 drivers is disturbingly sparse, raising a question about the interpretability of such data. Presumably because of this limitation, results of RNAi against sex-determining genes are often presented only when such manipulation results in reproduction of phenotypes in systemic mutants (that is, when the result is interpretable). Considering that cell-autonomous role of *dsx* or *fru* is not necessarily the focus of our study, we do not think either RNAi or CRISPRi will further support our conclusions.

MARCM can circumvent the problem of timing and strength because the mutations become homozygous at the cell division. However, MARCM also has an often-overlooked weakness that any cell clones in which a GAL4 driver is not active remain undetected. Moreover, creation of animals with mutant cells only in the desired neuronal population is exceedingly difficult, often requiring hundreds of individuals to identify a handful of suitable samples. Because the distribution of cell clones can be analyzed only after behavioral experiments, and since there is substantial inherent variability of behavior, an unreasonable amount of labor would be required to obtain interpretable data using MARCM.

We are aware of the limitation of constitutive mutants, but systematic analysis of such mutants is fundamental to interpret cell-specific manipulations of sex-determining genes. We hope reviewers will understand the value in what we can observe, and allow us to address functions of *dsx* or *fru* within specific neuronal populations in future studies.

Subsection “fruM specifies aggression-promoting Tk-GAL4FruM neurons”:The logic of transition from P1/pC1 to Tk-Gal4 neurons is very difficult to follow. The relationship between the former and the latter need to be explained. Are they completely different stories?

As mentioned at the beginning of this letter, we have split the original submission into 2 parts. This point was also raised by reviewer #3, and we hope the current format better conveys our central messages.

Figure 6:As a control, the authors should demonstrate that Tk-GAL4 neurons do not express dsx.

Please see Figure 1—figure supplement 2C.

Subsection “Differential roles of fru isoforms on male-male interactions and specification of Tk-GAL4FruM neurons”:The authors showed that the Tk-Gal4 neurons exist in fru∆B mutant males but behavior is affected in these flies. The simplest explanation would be that the neurites are malformed in these flies.Although the authors mentioned that 'fruMA and fruM mutants did not affect […] their arborization pattern', they didn't seem to carry out any quantitative morphological analyses. Such analyses should be included.Subsection “The Tk-GAL4FruM neurons in fruMB mutants can induce male aggression”:The authors mention that 'these mutants have Tk-GAL4^FruM^ neurons that appear to retain their morphology'. But how did they examine this?

We now show the averaged images of Tk-GAL4^FruM^ neurons in fruMA and fruMB mutants, which is in fact very similar to those in wild type males (“Layered roles of *fruitless* isoforms in specification and function of male-type aggression-promoting neurons in *Drosophila*”, Figure 5F-H, Video 7). The volume quantification of three major neural processes in these three genotypes indeed shows that they are indistinguishable from each other.

Again, this level of analysis may not reveal subcellular differences in neuromorphology among regular males, fruMA mutants, and fruMB mutants. However, we would like to point out that our conclusion that fruMB does not affect the specification of Tk-GAL4^FruM^ neurons is based on physiological and behavioral data, as well as neuroanatomical data (“Layered roles of *fruitless* isoforms in specification and function of male-type aggression-promoting neurons in *Drosophila*”, Figure 6 and Figure 5, respectively). The fact that optogenetic stimulation of Tk-GAL4^FruM^ neurons in fruMB mutants triggers aggression as robustly as in regular males suggests that the overall integrity of Tk-GAL4^FruM^ neurons remains largely unaffected in fruMB mutants.

Reviewer #3:This paper is a paradox. As far as I can tell it is rigorous on all levels. The scholarship is excellent, deep and well rounded. In addition, the issues are explored thoroughly and there is an enormous amount of data. In addition, an enormous number of fly lines (the table that describes the lines is phenomenal). But on the other hand, it is too much. I would like to discuss the argument and conclusions that surface relating dsx and fru expressing neurons with sexually dimorphic choice vs. sexually dimorphic motor patterns and the interesting differences between reproductive behavior and aggression, but the paper is simply too dense at almost every point. Reading it is a painful experience and this does the work a tremendous disservice. I say this recognizing and admiring the care that has gone into preparing this text. It seems flawless, just impossibly dense.

We understand a reservation that echoes with other reviewers’ concerns regarding the clarity of our original submission. We would like to address the reviewer’s specific suggestions in the following sections.

What to do? Here is a list of suggestions.1) The text and figures are impeccably constructed but for the average reader they are overly complicated and far too dense. The flow of the manuscript would benefit greatly from severe editing of the text, figures and figure legends. It is with a heavy heart that I ask the authors to do this knowing the amount of effort it took to construct the manuscript.

We significantly reorganized our original submission. Namely, we first understood the value of the last suggestion and split the original version into two new manuscripts as we already described. We hope this will substantially improve the clarity of our messages.

2) Remove raster plots from figures throughout and move to supplemental material if needed. See Figure 1, Figure 4, Figure 5, Figure 6, Figure 7, Figure 9.

We moved the raster plots of behaviors largely to figure supplement panels.

3) Figure 2 is not required. Move to supplemental material and concisely state result in text.

We feel the evidence suggesting that P1^a^ and NP2631 ∩ dsx^FLP^ neurons are separate populations has significance, especially now that we detected different levels of contribution of *fru* on these two neuronal populations. We therefore decided to keep the panels in “Sex-determining genes distinctly regulate courtship capability and preference toward females through sexually dimorphic neurons”, Figure 5.

4) Much of the immunohistochemistry could be removed from figures and moved to supplemental material.

*See Figure 3, Figure 6, Figure 8.*This is in contrast to requirements from reviewers #1 and #2 to perform more rigorous neuroanatomical analyses. We followed a custom for most *Drosophila* papers and kept representative images, but we would be open to further suggestions for improving clarity.

5) The summary panels at the end of some of the figures are helpful. But a single unifying summary/model would be best. Though organized in a systematic and rational way the results still feel piecemeal and it is tiresome to continually have refer back to previous figures or supplemental material to regain your bearing. A single model would help immensely.

We reduced the number of models to one for “Sex-determining genes distinctly regulate courtship capability and preference toward females through sexually dimorphic neurons” (Figure 4D), and two for “Layered roles of *fruitless* isoforms in specification and function of male-type aggression-promoting neurons in *Drosophila*” (Figure 2D and Figure 6H). It was a challenge to summarize our findings in a single model, given that *dsx* and *fru* have different roles on 3 populations of neurons and 2 types of social behaviors. We hope the new models make our core finding clear.

[Editors’ note: what follows is the authors’ response to the second round of review.]

Essential revisions:The issues enumerated below need to be clarified before the paper can be accepted.1) That activation of Tk-Gal4 neurons in single-housed FruM females does not elicit aggression is at odds with the conclusion that this circuit is sufficient for executing male type aggression. If anything, single housing should have increased aggression. The authors' suggest that the lack of aggression is because of increase in male-type courtship, but this still tacitly indicates that activating the Tk/FruM circuit is insufficient to induce aggression. The authors' conclusions should be tempered accordingly.

We appreciate the reviewer’s attention to point out an imprecise choice of words (“sufficient”). We rephrased the sentence in “Layered roles of *fruitless* isoforms in specification and function of male-type aggression-promoting neurons in *Drosophila*”, subsection “Female-type NP2631 ∩ dsx^FLP^ neurons in fruM females do not promote aggression**”** to more accurately convey our point, as detailed below.

Our conclusion in the original manuscript that the fruM-dependent circuit (which includes Tk-GAL4^FruM^ neurons) is “sufficient for executing male-type aggression” does not mean that the presence and/or the activation of Tk-GAL4^FruM^ neurons and its downstream is “sufficient” to recapitulate all aspects of male-type aggressive behaviors. What we wanted to express is that the activity of Tk-GAL4^FruM^ neurons can generate male-type lunges in the absence of neural components specified by *dsx*. Although the importance of fruM on organizing male-type aggressive behavior was demonstrated in Vrontou et al., 2006, our point is noteworthy because not all fruM-expressing neurons are required for the animals to perform a lunge. Male-type NP2631 ∩ dsx^FLP^ neurons and P1^a^ neurons are such “non-essential” *fru*-expressing neuronal components (“Layered roles of *fruitless* isoforms in specification and function of male-type aggression-promoting neurons in *Drosophila*”, subsection “fruM specifies aggression-promoting Tk-GAL4^FruM^ neurons”: see also “Sex-determining genes distinctly regulate courtship capability and preference toward females through sexually dimorphic neurons”, Figure 4), even though their activity can also promote lunges in a (normal) male. As we discussed in “Sex-determining genes distinctly regulate courtship capability and preference toward females through sexually dimorphic neurons”, *dsx* plays an important role in specifying both neuronal populations.

We showed in Figure 1C and D that activation of Tk-GAL4^FruM^ neurons did not elicit aggression toward female target flies (which is consistent with the paper that first described Tk-GAL4^FruM^ neurons for male aggression (Asahina et al., 2014)). To us, the observation that artificial activation of a neuronal population can have different behavioral impacts in a context-dependent manner is an important conclusion (see also subsection “Sex of the target animals is an important biological variable” in the Discussion of “Sex-determining genes distinctly regulate courtship capability and preference toward females through sexually dimorphic neurons”). In fact, our observation that fruM females can attack female targets when their Tk-GAL4^FruM^ neurons were activated suggests that the absence of lunges from a male tester to a female target may be at least partially dependent on the function of *dsx* (Figure 2D). In this sense, the lack of “sufficiency” that the reviewer pointed out is inherent to the function of Tk-GAL4^FruM^ neurons itself (as stated in subsection “fruM specifies aggression-promoting Tk-GAL4^FruM^ neurons”), and not necessarily an issue specific to fruM females.

2) As with the companion paper (“Sex-determining genes distinctly regulate courtship capability and preference toward females through sexually dimorphic neurons”), it would appear that the effects of Fru isoform on aggression in Tk-GAL4FruM neurons also can only really be assessed by looking at animals were the Tk-GAL4FruM neurons are mutant/perturbed for fru function (e.g. fru isoform-specific mutant MARCM clones, UAS-Cas9 system, or fru-RNAi). Otherwise, the data would need to be interpreted with the caveat that fru mutant phenotypes could be caused by lack of Fru isoform in other neurons in the CNS and/or the PNS. This issue needs to addressed directly – ideally experimentally given that a lineTK-GAL4/fruP1LexA/LexAop2-FLPL that may allow these neurons to be targeted has been described in Asahina et al., 2014.

This is a parallel critique to our Ishii et al., (2020). While we acknowledge that the molecular mechanism by which *dsx* and *fru* specified neural sexual dimorphism is an important question, note that “the effects of Fru isoforms on aggression in Tk-GAL4^FruM^ neurons” was not a question we aimed to address in this manuscript. Please see our response to a parallel question for “Sex-determining genes distinctly regulate courtship capability and preference toward females through sexually dimorphic neurons”, which includes the discussion why the interpretation of RNAi or dCas9-mediated gene knockdown (which depends on the GAL4 expression), as well as MARCM analysis, have caveats for studying genes involved in development. We argue that morphological and functional phenotypes of the neurons of interest in a systemic mutant is an important knowledge that deserves attention on its own, precisely because of challenges in addressing functions of these genes within specific cell types.

We would like to point out that Tk-GAL4^FruM^ neurons present an extreme case of this challenge. Since Tk-GAL4^FruM^ neurons are not labeled in the absence of fruM, one likely scenario is that fruM is required cell-autonomously. Neither RNAi or dCas9 would be a valid method to address this possibility because GAL4 must turn on for RNAi or dCas9 to work. This is a classic “catch-22” situation: the very fact that GAL4 turns on likely means that the fate of the cell is already determined before RNAi or dCas9-mediated knockdown of *fru* has an opportunity to influence its developmental process. In other words, there is no possible way for RNAi or dCas9 to suppress the activity of GAL4 that drives them if fruM is required cell-autonomously to specify Tk-GAL4^FruM^ neurons. If Tk-GAL4^FruM^ neurons disappear like in fruF males – it will be an even bigger mystery that RNAi or dCas9 can exert its effect without GAL4 activity.

It is certainly possible that *fru* is required outside of Tk-GAL4^FruM^ neurons to specify its sexual dimorphism, but results of RNAi or dCas9 (that is, that Tk-GAL4^FruM^ neurons appear the same as the genetic controls) would not be interpretable. In other words, both RNAi and dCas9 are fundamentally flawed approaches to adequately address this important question.

3) A key finding arising from optogenetic experiments, artificially activating Tk-GAL4FruM neurons are a refinement of the authors previously published thermogenetic work (Asahina et al., 2014), and the isoform-specific roles of Fru on male behavior and Fru-specified neural circuits, are implied by work from other groups (Neville et al., 2014; von Philipsborn et al., 2014). The discussion must clearly articulate what new information thais manuscript provides on Fru isoform function and aggressive behavior.

We would like to emphasize that the previous publication never addressed Tk-GAL4^FruM^ neurons’ specification (Figure 1H, K-P) or functions (Figure 2) in “fruM females”. Also, the contribution of specific *fru* isoforms on male-male aggression has never been addressed in either of the previous important publications (Neville et al., 2014; von Philipsborn et al., 2014). Therefore, the findings on the differential roles of *fru* isoforms for male aggression (Figure 4) are entirely new. Also, how the specification of Tk-GAL4^FruM^ neurons (Figure 5) and its function on aggressive behaviors (Figure 6) are affected by *fru* isoform mutants has never been tested. We therefore state that all key data presented here are novel, and not a mere “refinement of the previous publication (Asahina et al., 2014)” as the reviewer suggested.

Upon the reviewer’s suggestion, we clearly indicated that the courtship defects of fruMB and fruMC mutants were reported in the previous publications (“Layered roles of *fruitless* isoforms in specification and function of male-type aggression-promoting neurons in *Drosophila*”, subsections “Female-type NP2631 ∩ dsx^FLP^ neurons in fruM females do not promote aggression” and “Differential roles of *fru* isoforms on male-male interactions and specification of Tk- GAL4^FruM^ neurons”). Also note that we acknowledged *fru* isoforms’ functions on gene expression, neuroanatomy, and behavior have been investigated in the previous publications (“Layered roles of *fruitless* isoforms in specification and function of male-type aggression-promoting neurons in *Drosophila*”, Discussion section). I hope these clarifications resolve any confusion about what has been known previously. We performed courtship assays on isoform-specific mutants (Figure 4—figure supplement 1A, B) as a control for our genetic manipulations, which we felt was important for demonstrating that fruMB and fruMC mutants are qualitatively different even though their phenotype on aggression (both strong reduction) appears similar.

4) Does Figure 2—figure supplement 2E, F1 (subsection “Tk-GAL4FruM neurons are a part of a fruM-dependent circuit for the execution of male-type aggressive behaviour”) refer to a different figure panel. Please correct.

Thank you for pointing out this mistake. It was supposed to refer to Figure 2—figure supplement 3E, F_1_, and was corrected accordingly.

5) Dsx immunolabeling in Figure 1—figure supplement 2C is poor. Please provide better images.

Please see the new Figure 1—figure supplement 2C and D for new images, which was generated with a new batch of anti-DsxM and anti-FruM antibodies. As stated in the other letter, these new antibodies coupled with a modified fixation process clearly showed that Tk-GAL4^FruM^ neurons indeed express FruM, but not DsxM.